# Reducing lipid bilayer stress by monounsaturated fatty acids protects renal proximal tubules in diabetes

**Albert Pérez-Martí[1], Suresh Ramakrishnan[1], Jiayi Li[1], Aurelien Dugourd[2], Martijn R Molenaar[3], Luigi R De La Motte[1], Kelli Grand[4], Anis Mansouri[2], Mélanie Parisot[5], Soeren S Lienkamp[4], Julio Saez-Rodriguez[2,6], Matias Simons[1,6]***

[1]Section Nephrogenetics, Institute of Human Genetics, University Hospital Heidelberg, Heidelberg, Germany; [2]Institute for Computational Biomedicine, Faculty of Medicine, Heidelberg University, University Hospital Heidelberg, Heidelberg, Germany; [3]European Molecular Biology Laboratorium (EMBL), Structural and Computational Biology Unit, Heidelberg, Germany; [4]Institute of Anatomy, University of Zurich, Zurich, Switzerland; [5]Genomics Core Facility, Institut Imagine-Structure Fédérative de Recherche Necker, INSERM U1163, INSERM US24/CNRS UMS3633, Paris Descartes Sorbonne Paris Cite University, Paris, France; [6]Molecular Medicine Partnership Unit (MMPU), European Molecular Biology Laboratory (EMBL) and Heidelberg University, Heidelberg, Germany

**Abstract** In diabetic patients, dyslipidemia frequently contributes to organ damage such as diabetic kidney disease (DKD). Dyslipidemia is associated with both excessive deposition of triacylglycerol (TAG) in lipid droplets (LDs) and lipotoxicity. Yet, it is unclear how these two effects correlate with each other in the kidney and how they are influenced by dietary patterns. By using a diabetes mouse model, we find here that high-fat diet enriched in the monounsaturated oleic acid (OA) caused more lipid storage in LDs in renal proximal tubular cells (PTCs) but less tubular damage than a corresponding butter diet with the saturated palmitic acid (PA). This effect was particularly evident in S2/S3 but not S1 segments of the proximal tubule. Combining transcriptomics, lipidomics, and functional studies, we identify endoplasmic reticulum (ER) stress as the main cause of PA-induced PTC injury. Mechanistically, ER stress is caused by elevated levels of saturated TAG precursors, reduced LD formation, and, consequently, higher membrane order in the ER. Simultaneous addition of OA rescues the cytotoxic effects by normalizing membrane order and increasing both TAG and LD formation. Our study thus emphasizes the importance of monounsaturated fatty acids for the dietary management of DKD by preventing lipid bilayer stress in the ER and promoting TAG and LD formation in PTCs.

***For correspondence:**
matias.simons@med.uni-heidelberg.de

## Editor's evaluation

This study addresses the differential effect of saturated versus monounsaturated lipids on damage to kidney tubules. The authors find that the majority of the lipids are taken up in the S2 and S3 segments of the proximal tubule. They find that while the unsaturated fatty acid increased lipid storage in the tubule cells, it produced less cell damage than the saturated fatty acid due to decreased endoplasmic reticulum stress. These studies emphasize a potential beneficial effect of a diet rich in monounsaturated fatty acids, especially in patients with chronic kidney disease.

## Introduction

Diabetic kidney disease (DKD; or diabetic nephropathy) is the most common complication of diabetes mellitus defined as diabetes with albuminuria or an impaired glomerular filtration rate (GFR) or both. It is the leading cause of end-stage renal disease, necessitating dialysis or transplantation. Despite improved management of diabetes, the number of DKD patients continues to rise, causing an enormous health and economic burden worldwide. Classical histopathological features of DKD are glomerular changes such as podocyte hypertrophy and loss, glomerular basement membrane thickening, mesangial expansion, and Kimmelstiel–Wilson nodules (*Oshima et al., 2021*). Often overlooked are the tubulointerstitial alterations, including peritubular fibrosis, that contribute to or even drive DKD progression (*Bonventre, 2012*). The recent success of sodium-glucose co-transporter-2 (SGLT2) inhibitors highlights the proximal tubule cell (PTC) as an important target in DKD therapy (*DeFronzo et al., 2021*).

Constituting more than half of renal mass (*Park et al., 2018*), PTCs reabsorb most of the solutes and proteins filtered by the glomerulus (*Chevalier, 2016*). While solute transport is carried out by dedicated transporters using ion gradients established by the $Na^+/K^+$-ATPase, protein uptake occurs via an endocytic machinery with low cargo specificity and unusually high capacity. The energy for both tasks is almost exclusively provided by mitochondrial β-oxidation of fatty acids (*Bobulescu, 2010*; *Kang et al., 2015*). The fatty acids are taken up from the blood side or by albumin that is partially filtered by the glomerulus and delivers bound fatty acids from the luminal side to the PTCs. Accordingly, in mice fed with a high-fat diet (HFD) or in individuals with type 2 diabetes, lipids accumulate predominantly in PTCs (*Herman-Edelstein et al., 2014*; *Kang et al., 2015*; *Rampanelli et al., 2018*), indicating that these cells may be equipped with a high capacity to take up and store lipids. Lipid overabundance, however, can lead to 'lipotoxicity,' which is a main driver of kidney disease progression (*Abbate et al., 2006*; *Bobulescu, 2010*; *Moorhead et al., 1982*). This is particularly true in DKD, where albuminuria combined with dyslipidemia leads to a tubular overload of albumin-bound fatty acids (*Bonventre, 2012*; *Zeni et al., 2017*).

Within cells, lipid overabundance leads to enhanced triacylglycerol (TAG) synthesis and lipid droplet (LD) formation. This process occurs at the endoplasmic reticulum (ER), where three fatty acids are consecutively added via esterification to the glycerol backbone beginning with *sn*-glycerol-3-phosphate. Once enough TAGs (and cholesterol esters) have been deposited between the ER lipid bilayer leaflets, LDs bud into the cytoplasm enwrapped by a phospholipid monolayer (*Wilfling et al., 2014*). In adipocytes, which are specialized in lipid storage, this is a physiological process. In other cell types, excessive LD formation is often a sign of impaired cellular homeostasis. A well-known example is hepatic steatosis that is featured by increased lipid storage in LDs of hepatocytes and often progresses towards liver fibrosis (*Seebacher et al., 2020*). Also in type 2 diabetes, lipid accumulation is a common feature in many organs contributing to insulin resistance. However, as free fatty acids (in particular, saturated ones) can activate pro-inflammatory pathways (*Shi et al., 2006*) or generate reactive oxygen species (ROS) upon excessive mitochondrial ß-oxidation (*Gehrmann et al., 2015*), storing fatty acids in LDs could also prevent damage (*Listenberger et al., 2003*). Accordingly, it has been shown in neural stem cells that LDs can sequester polyunsaturated acyl chains, protecting them from the oxidative chain reactions that generate toxic peroxidated species and ferroptosis (*Bailey et al., 2015*; *Dierge et al., 2021*). Similarly, sequestering free fatty acids by LDs has been shown to protect mitochondrial function during starvation-induced autophagy (*Nguyen et al., 2017*). Therefore, LDs can be damaging or protective depending on the tissue context. In the context of DKD, however, there is only limited knowledge of the role of TAG and LD formation.

In this study, we treated hyperglycemic mice with two different HFDs, one enriched in butter (containing high amounts of palmitic acid [PA]) and one enriched in olive oil (containing high amounts of oleic acid [OA]). While the butter diet caused more tubular damage and renal fibrosis than the olive oil, less lipid accumulation was observed in the renal proximal tubules. The tubular damage was more pronounced in PTCs showing less LDs. On a wider level, the integration of lipidomic, transcriptomic, and functional studies revealed that PA induced rapid cytotoxicity by increasing the relative proportion of disaturated TAG precursors in cellular membranes. This leads to ER stress, which can be fully suppressed by co-incubating with OA. This protective effect is tightly connected with the formation of unsaturated phospholipids and the formation of LD that serve as a lipid reservoir to protect against lipid bilayer stress in the ER.

## Results

### An HFD enriched in saturated fatty acids causes tubular LD accumulation and kidney damage in diabetic mice

We wanted to study the effect of overloading PTCs with saturated and unsaturated fatty acids (SFA) for DKD progression in mice. For this, we combined a low-dose streptozotocin (STZ) regimen, which destroys β-pancreatic islets and produces insulin deficiency, with two types of HFD enriched in SFA and monounsaturated fatty acids (MUFA). Both HFDs contained 20 kcal% of protein, 35 kcal% of carbohydrates, and 45 kcal% of fat, whereas the control diet contained 24 kcal% of protein, 58 kcal% of carbohydrates, and 18 kcal% of fat. The source of fat was butter for the SFA-HFD, olive oil for the MUFA-HFD, and the standard soybean oil for the control diet (see also *Appendix 1—table 1*). The different diets were started at 7 weeks of age, and mice were followed for 20 weeks in four groups: while mice on control diet, SFA-HFD and MUFA-HFD received five consecutive daily injections of STZ (50 mg/kg) at 11 weeks of age, one group of mice on control diet was injected with the vehicle. Nonfasting hyperglycemia became apparent 4 weeks after STZ injection (*Figure 1—figure supplement 1A*). In addition, STZ-injected mice presented polyphagia and polydipsia as a sign of increased glucosuria (*Figure 1—figure supplement 1B and C* and *Appendix 1—table 2*). In addition, they showed increased levels of albumin in the urine (*Appendix 1—table 2*).

All three STZ-injected groups showed impaired weight gain over the entire experiment compared to vehicle-injected mice. Among the STZ-injected groups, no differences in body weight were observed except for the last two weeks when the mice fed a control diet decreased weight (*Figure 1A*). All STZ-injected mice had an extreme loss of epididymal white adipose tissue (eWAT), with some mice losing all their eWAT (*Figure 1B*). Concomitantly, plasma TAGs levels were increased in diabetic mice fed with MUFA-HFD and SFA-HFD (*Figure 1C*). Liver weight was only increased in mice fed with the two HFDs while all diabetic groups showed increased kidney weight (*Figure 1B*). We used periodic acid-Schiff (PAS) and Oil Red O (ORO) staining to determine whether the increase in kidney weight was due to ectopic fat deposition. Diabetic mice on both HFDs massively accumulated LDs in the kidney cortex. In the MUFA-HFD group, the percentage of kidney cortex stained with ORO was higher than in the SFA-HFD group. Individual LDs could be seen in tubules but not in glomeruli (*Figure 1D–F*, *Figure 1—figure supplement 2A*). To determine what tubular segments accumulated the lipids, LDs were labeled using BODIPY, a neutral lipid staining dye, while PTC segments were marked with either megalin for all three PTC segments (*Chen et al., 2019*) or SGLT2 for only the S1 segment (*Vallon et al., 2011*). Lipid accumulation was particularly evident in PTCs of the straight S2 or S3 segments but not of the convoluted S1 segment (*Figure 1E*, *Figure 1—figure supplement 2B*).

To assess whether STZ and HFD treatments caused renal damage, we first studied circulating levels of the tubular damage marker LCN2. At 12 weeks after the STZ injection, LCN2 levels were higher in mice fed with SFA-HFD than in mice fed with MUFA-HFD (*Figure 1G*). To determine whether tubular damage resulted in fibrosis, we made use of Picro-Sirius Red staining to visualize collagen deposition. The staining revealed fibrotic areas in all diabetic mice groups, but only in the mice fed SFA-HFD this increase reached statistical significance (*Figure 1H and I*). Moreover, when normalized to fat deposition, the increase in the fibrotic area became even more apparent in SFA-HFD kidneys compared to MUFA-HFD kidneys (*Figure 1J*). Increased mRNA levels of the profibrotic molecules *Ccl5* and *Fn1* in SFA-HFD kidneys supported the histological findings (*Figure 1K*). Finally, we performed immunohistochemistry to detect changes in the expression of the tubular damage marker KIM-1 (*Mori et al., 2021*). We found strong apical KIM-1 expression in renal proximal tubules in some of the STZ and SFA-HFD mice while only moderate in MUFA-HFD kidneys (*Figure 1—figure supplement 2D*).

Taken together, the STZ-induced diabetic mice recapitulated diabetic features, namely, hyperglycemia, polyphagia, polydipsia, polyuria, glucosuria, albuminuria, and lipodystrophy. While the MUFA-HFD diet favored the accumulation of LDs in the S2/S3 segments of the renal proximal tubules, the overload with SFAs resulted in higher tubular damage and fibrosis.

### PA impairs cell viability in PTC culture models that can be rescued by OA

To better understand the molecular mechanisms driving SFA-mediated renal damage as well as MUFA-mediated renoprotection, we used induced renal epithelial cells (iRECs), which are proximal-tubule

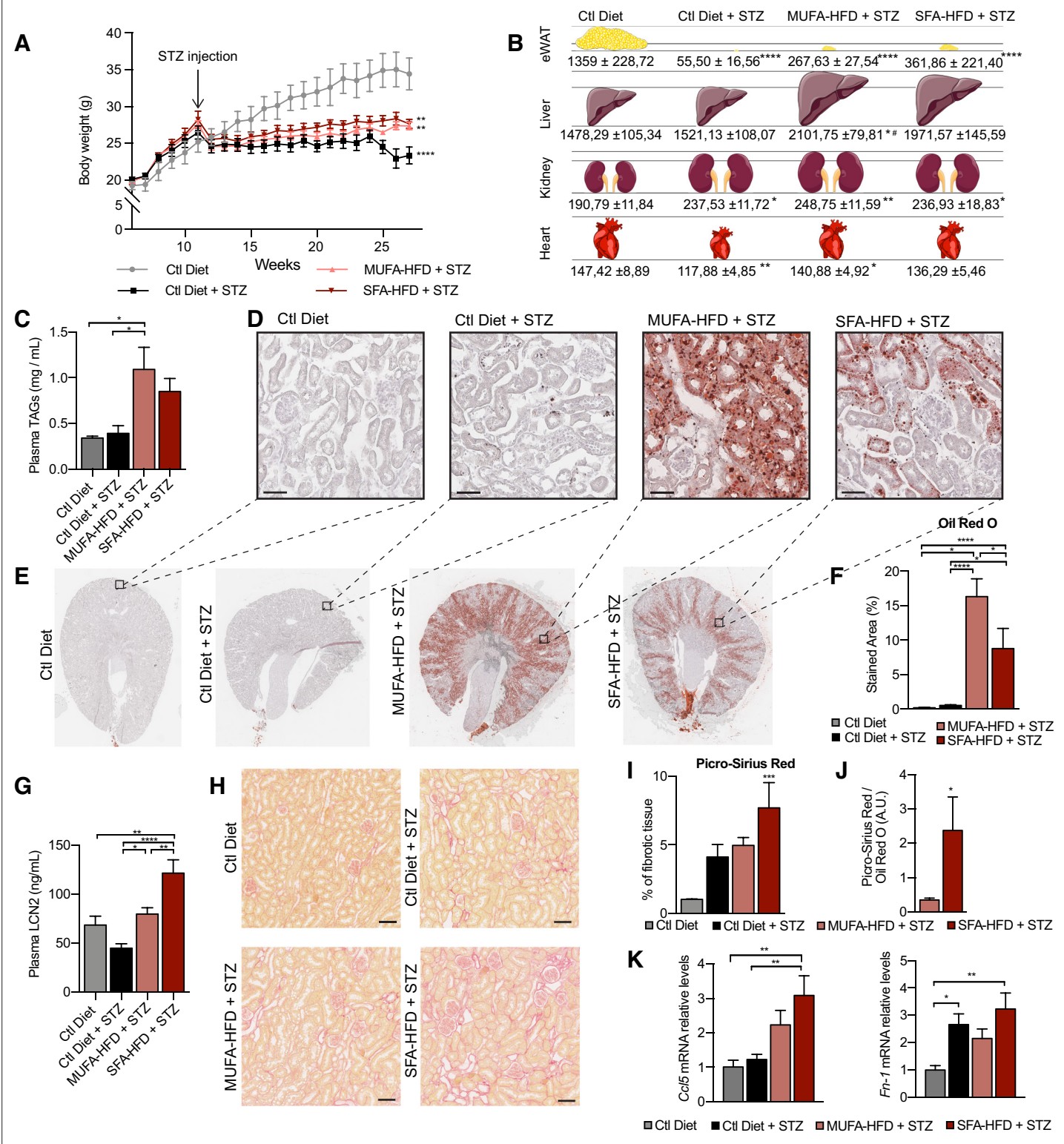

**Figure 1.** Saturated fatty acids high-fat diet (SFA-HFD) induces more tubular damage than monounsaturated fatty acids high-fat diet (MUFA-HFD) despite lower fat accumulation in mice. (**A**) Mice body weight throughout the experiment. The X-axis indicates the age of mice. (**B**) Schematic representation of tissue weight. (**C**) Plasmatic triacylglycerol (TAG) levels at week 16 after streptozotocin (STZ) injection. (**D–F**) Representative bright-field images of whole kidney sections (**E**) and cortex magnification (**D**) stained with Oil Red O (ORO). Quantification of the stained cortex area (**F**). Scale bars: 50 µm. (**G**) Plasmatic LCN2 levels at week 12 after STZ injection. (**H, I**) Representative bright-field images of kidney cortex stained with Picro-Sirius Red (**H**) and the quantification of the fibrotic cortex area (**I**). Scale bars: 50 µm. (**J**) Fibrotic area detected by Picro-Sirius Red normalized by fat deposition

*Figure 1 continued on next page*

*Figure 1 continued*

measured by ORO in mouse kidney cortex. (**K**) Quantitative RT-PCR detection of *Ccl5* and *Fn-1* expression levels in mouse kidney cortex. In (**A, C, G, I–K**), data are presented as mean ± SEM. *p<0.05, **p<0.01, ***p<0.001, ****p<0.0001; one-way ANOVA plus Holm–Sidak's multiple comparisons test. In (**B**), data are presented as mean ± standard deviation. *p<0.05, **p<0.01, ****p<0.0001 vs. Ctl; #p<0.05 vs. Ctl diet + STZ; one-way ANOVA plus Holm–Sidak's multiple comparisons test. (**A–C, F, G, I, J, K**), n = 7Ctl diet, n = 8Ctl diet + STZ, n = 8 MUFA-HFD+ STZ, n = 7 SFA-HFD+ STZ.

The online version of this article includes the following figure supplement(s) for figure 1:

**Figure supplement 1.** Diabetic phenotypes in streptozotocin (STZ)-injected mice.

**Figure supplement 2.** Lipid droplets and KIM-1 expression in kidneys of diabetic mice.

like cells directly programmed from fibroblasts (*Kaminski et al., 2016*). First, iRECs were treated with increasing concentrations of fatty acid-free bovine serum albumin (BSA) to determine the toxicity of albumin itself. However, none of the concentrations of BSA induced any cytotoxicity as measured by the uptake of Cytotox dye using the IncuCyte Live-Cell analysis system (*Figure 2—figure supplement 1A*). With this approach, we assessed the cytotoxicity in iRECs treated with PA, OA, and combinations of both. PA treatment induced cell death after 8 hr in a concentration-dependent manner, while cell viability was unaffected by OA, even after treatments of up to 7 days (*Figure 2A*, *Figure 2—figure supplement 1B*). Both PA and OA effects were independent from the level of confluency (*Figure 2B and C*). Importantly, co-treatment with OA completely rescued the cytotoxic effects of PA. Even when only 0.25 mM OA was co-incubated with 0.5 mM PA, cell viability was fully restored (*Figure 2A–C*). To more accurately reflect the diabetic environment, we also treated cells with BSA-PA in media containing increasing concentration of glucose. Increasing glucose levels mildly enhanced the cytotoxic effects of PA, but this was not statistically significant (*Figure 2—figure supplement 1C*). To compare the effects with other PT cell culture models besides iRECs, we finally performed the same cytotoxicity assays on murine primary PT cultures as well as OK cells. In both cell models, comparable effects on cell viability were observed when exposed to different doses of albumin fatty acids (*Figure 2—figure supplement 1D and E*). Altogether, it can be concluded that in cultured PTCs PA treatment rapidly leads to cytotoxic effects that can be rescued by OA.

## PA-induced cell injury elicits a unique transcriptional response

Next, we performed a comparative transcriptomic study on BSA-, BSA-PA-, BSA-OA-, and BSA-PA/OA-treated iRECs using RNA sequencing. The differential expression analysis showed that the transcriptional response observed in BSA-PA and BSA-OA cells was much more pronounced than in BSA-PA/OA-treated cells (*Figure 2D–F*). As we were particularly interested in biological processes that were up- or downregulated by the addition of PA and reverted to normal levels when cells were co-incubated with OA, we clustered genes accordingly and applied Gene Ontology (GO) analysis. Cluster 1, which included genes upregulated by PA and normalized by OA co-treatment, proved to be strongly enriched in genes involved in oxidative stress, ER stress, and autophagy (*Figure 2G*). Important lipid metabolism genes such as the mitochondrial fatty acid importer *Cpt1* and the fatty acid desaturases *Scd1* and *Scd2* also belonged to the top-regulated genes (*Figure 2H*). By contrast, cluster 4, featured by genes that were downregulated by PA and normalized by OA co-treatment, showed a clear enrichment of biological processes controlling cell proliferation (*Figure 2I*), confirming the observation that BSA-PA treatment slows down cell growth compared to the other three conditions (*Figure 2—figure supplement 1F*).

With the aim of identifying what transcription factors (TFs) might be regulating these cellular responses, we mined our transcriptomic data using DoRothEA (*Holland et al., 2020*), a comprehensive resource containing a curated collection of TFs and its transcriptional targets (see Materials and methods). We obtained TFs whose predicted target genes associated with one or more of the five different conditions (PA vs. control, PA vs. PA/OA, PA/OA vs. control, OA vs. control, OA vs. PA/OA). Interestingly, several of the identified TFs whose activities were upregulated by PA and downregulated by OA (*Figure 2—figure supplement 2*) have known roles in lipid metabolism (e.g., HNF4A, SREBF1) and ER stress (e.g., ATF4, ATF6). By contrast, several TFs whose activities were regulated in the opposite way function in cell cycle control (e.g., E2F4, MYC, FOXM1). Thus, our analysis identified TF candidates that may mediate effects on ER stress, lipid metabolism, and cell cycle in response to PA.

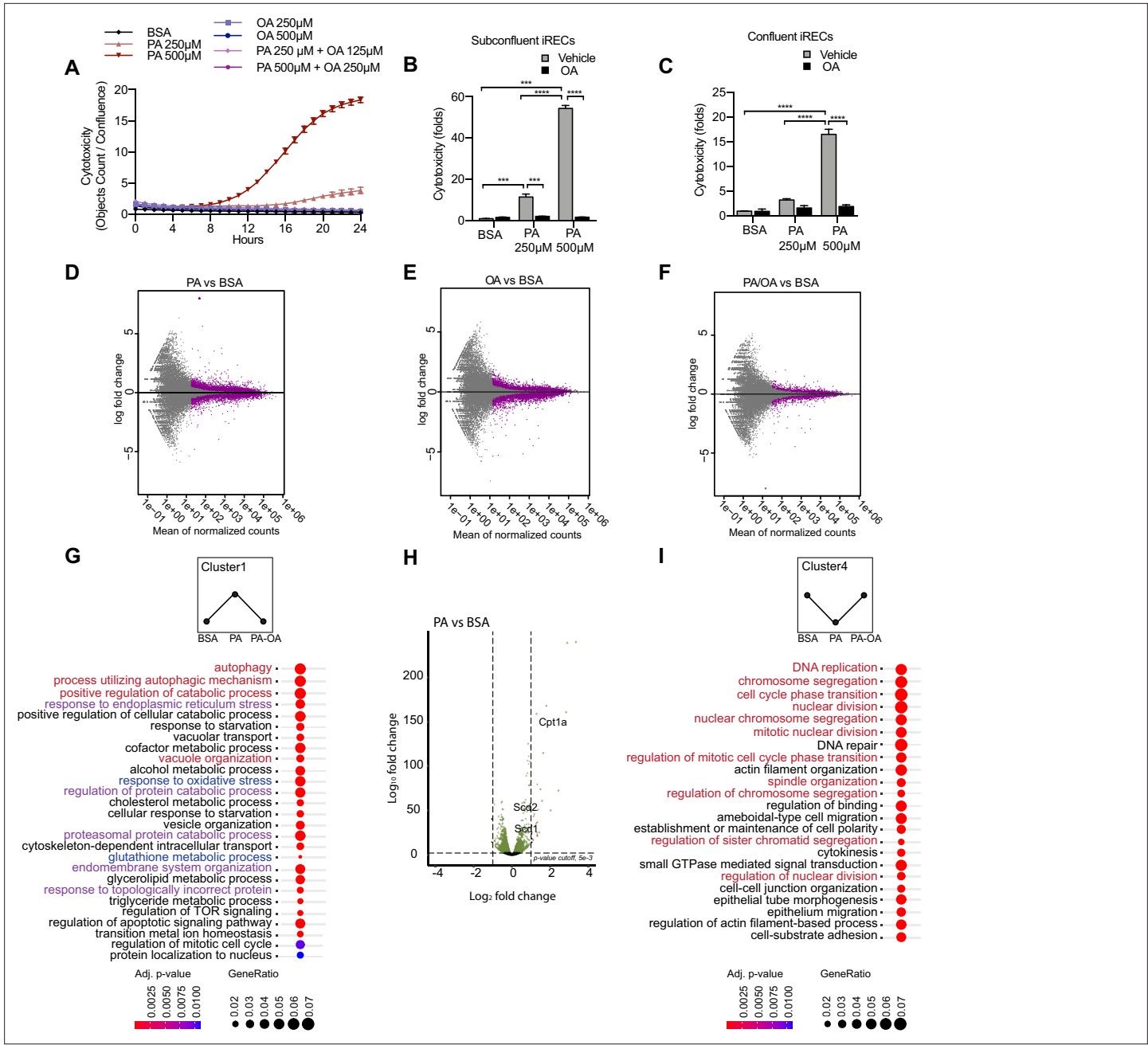

**Figure 2.** Palmitic acid (PA)-induced cytotoxicity is blocked by oleic acid (OA) in proximal tubular cells. (**A**) Cytotoxicity throughout 24 hr in induced renal epithelial cells (iRECs) treated with several combinations of bovine serum albumin (BSA)-bound fatty acids. (**B, C**) Cytotoxicity in subconfluent (**B**) and confluent (**C**) iRECs after 24 hr treatment with BSA, OA 250 µM, PA 250 µM, PA 250 µM + OA 125 µM, PA 500 µM, and PA 500 µM + OA 250 µM. Fold representation of objects counts/confluence. (**D–F**) MA plots of differentially expressed genes from iRECs treated for 16 hr with BSA, PA 250 µM, OA 250 µM, and PA 250 µM + OA 125 µM. Purple dots represent statistically significant changes. (**G, I**) Gene Ontology for biological processes overrepresentation analysis using clusterProfiler. The 25 most significant terms were plotted for clusters 1 (**G**) and 4 (**I**). The size of the spheres corresponds to the number of genes included and the color to the adjusted p-value. (**G**) Biological processes related to autophagy colored in red, related to oxidative stress in blue and related to endoplasmic reticulum (ER) stress in violet. (**I**) Biological processes related to cell proliferation colored in red. (**H**) Volcano plot of PA vs. BSA differential gene expression. Colored dots represent statistically significant changes. In (**A–C**), data are presented as mean ± SEM. *p<0.05, **p<0.01, ***p<0.001, ****p<0.0001; two-way ANOVA and Holm–Sidak's multiple comparisons test. (**D–I**) Significance was considered when adjusted p-value <0.05 (DESeq2 based on negative binomial distribution). (**A–I**) n = 3.

The online version of this article includes the following figure supplement(s) for figure 2:

**Figure supplement 1.** Characterization of factors contributing to lipotoxicity in iRECs and other proximal tubular cell (PTC) models.

*Figure 2 continued on next page*

*Figure 2 continued*

**Figure supplement 2.** Estimated transcription factor activity in response to palmitic acid (PA), oleic acid (OA), and PA/OA.

**Figure supplement 3.** Oleic acid (OA) suppresses palmitic acid (PA)-induced oxidative stress.

## PA-induced ROS formation is blocked by OA co-treatment

To functionally validate the transcriptional responses, we focused on the main cellular processes upregulated by PA (cluster 1). To study the oxidative stress response, we used dihydroethidium (DHE) to measure the generation of ROS. Superoxide levels were increased by PA treatment in a dose-dependent manner. When OA was added together with PA, the DHE staining returned to basal levels (*Figure 2—figure supplement 3A and B*). Next, we wondered whether the protective effect of OA on ROS formation was due to changes in mitochondrial activity. Using tetramethylrhodamine ethyl ester (TMRE), which is an indicator of mitochondrial membrane polarization, PA was found to increase mitochondrial activity while OA reduced it. The combination of PA with OA returned mitochondrial activity to basal levels (*Figure 2—figure supplement 3C and D*). As β-oxidation requires the import of fatty acids into the mitochondria by CPT1 (*Miguel et al., 2021*), which was upregulated by PA and OA, we used the CPT1 inhibitor etomoxir that was previously shown to decrease mitochondrial β-oxidation in iRECs (*Marchesin et al., 2019*). Etomoxir treatment increased the PA-induced cytotoxicity. However, OA was still protective in this condition (*Figure 2—figure supplement 3E*).

Together, these experiments showed that PA stimulates ROS generation and OA blocks it, validating the observed antioxidant response in the RNA-seq study. Since etomoxir boosted PA-induced cytotoxicity, increased mitochondrial fatty acid uptake and oxidation seem to be a beneficial response to PA overload. Yet, the protective effect of OA does not seem to involve these processes.

## Albumin PA triggers an ER stress response that can be rescued by OA

Next, we examined the role of ER stress in response to the PA insult. ER stress can be activated by three different branches known as the IRE1α, ATF6, and PERK branches (*Walter and Ron, 2011*). qPCR-based ER stress marker analysis of spliced *Xbp1* (*sXbp1*), *Hspa5* (also known as *Bip*), and *Ddit3* (also known as *Chop*) mRNA showed that all three ER stress response pathways are activated by PA after 16 hr. Full suppression of ER stress activation was achieved by cotreatment with OA (*Figure 3A*). In order to study which of the ER stress branches contributed most to cytotoxicity, we co-treated BSA-PA iRECs with inhibitors of PERK (GSK2606414), IRE1α (GSK2850163), and ATF6 (Ceapin-A7). On the one hand, the PERK inhibitor slightly reduced the cytotoxicity caused by PA, suggesting that the PERK-eif2a-ATF4 axis may contribute to the cell death process. On the other hand, the inhibition of IRE1α kinase and RNase activities exacerbated the PA cytotoxicity, arguing for a protective role of IRE1α in response to PA (*Figure 3B*).

As *Scd1* and *Scd2* were top-regulated genes in the BSA-PA condition, we also measured ER stress markers and cytotoxicity in iRECs treated with a low dose of PA plus the chemical SCD inhibitor CAY10566. The inhibitor exposed the ER stress responses and enhanced the cytotoxicity induced by low-dose PA, indicating that fatty acid desaturation is critical for preventing PA-induced ER stress (*Figure 3C and D*). We also measured ER stress markers in cells treated jointly with PA and etomoxir. Importantly, etomoxir exacerbated the ER stress response induced by PA, suggesting that decreased mitochondrial β-oxidation might increase the burden of PA in the ER (*Figure 3—figure supplement 1A*).

To investigate whether the protective effect of OA is specific for PA-induced ER stress, we treated BSA-OA cells with the chemical ER stressors tunicamycin and thapsigargin that are known to induce protein misfolding. Here, OA treatment did not show any protective effect against the ER stress caused by tunicamycin and thapsigargin (*Figure 3E*), suggesting that the protective effects of OA are specific for PA-induced ER lipid bilayer stress.

Finally, as prolonged ER stress can lead to the translocation of misfolded proteins into the cytoplasm via the ERAD system (*Lemberg and Strisovsky, 2021*), we reasoned that the increased expression of components of the autophagic machinery, such as p62, might be part of a proteostatic response. For functional validation, we first performed immunocytochemistry against the autophagy markers LC3 and p62. PA treatment mildly induced the formation of LC3 puncta as well as large perinuclear structures positive for p62, all of which were cleared by the addition of OA (*Figure 3—figure*

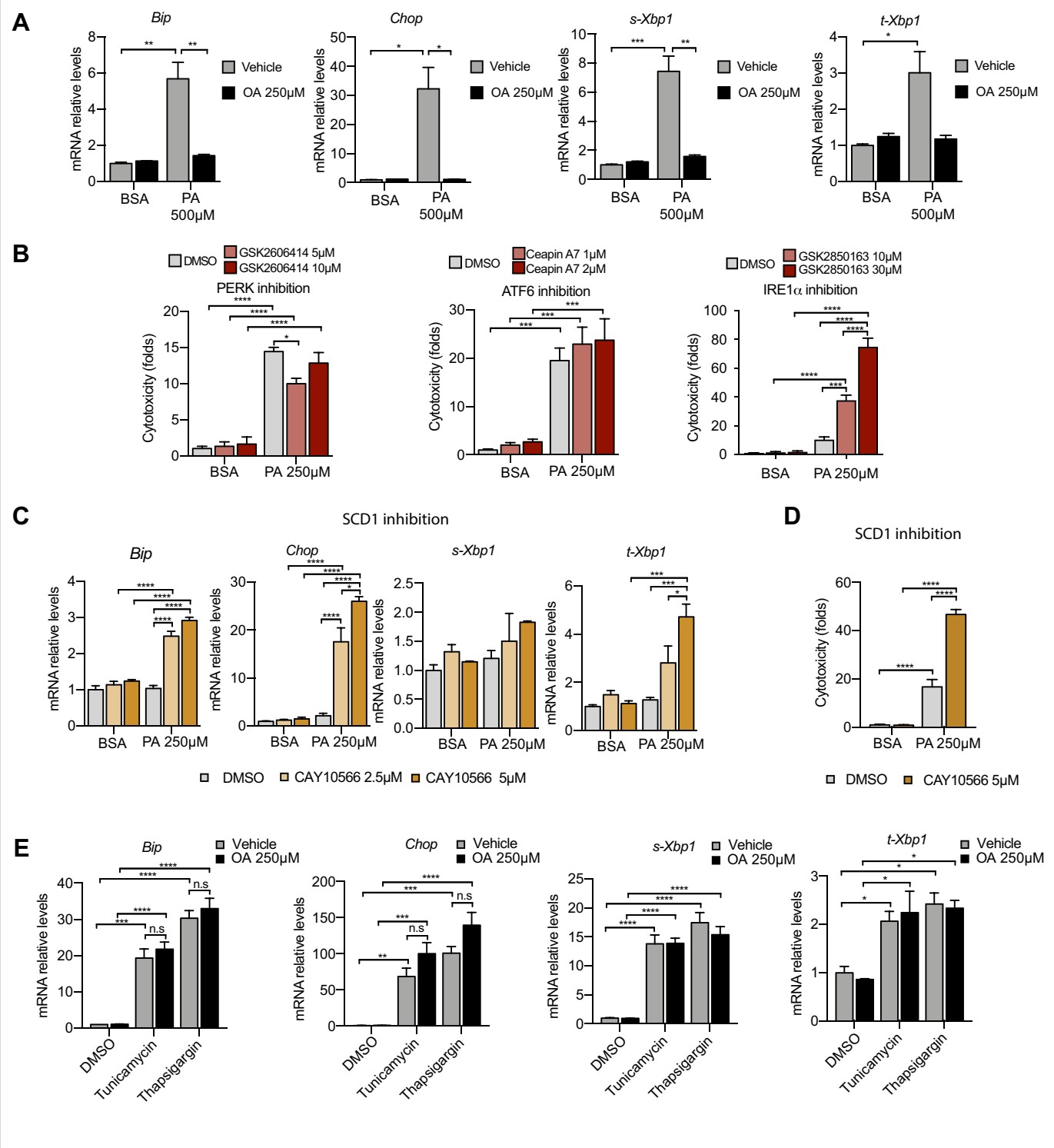

**Figure 3.** Excess of saturated fatty acids triggers the endoplasmic reticulum (ER) stress response. (**A, C, E**) Quantitative RT-PCR detection of ER stress markers in induced renal epithelial cells (iRECs) treated 16 hr with several combinations of BSA-bound fatty acids (**A**), palmitic acid (PA) plus ER stress signaling inhibitors (**B**), PA plus SCD1 inhibitor (**C**), and tunicamycin (10 μM) and thapsigargin (1 μM) plus oleic acid (OA) (**E**). (**B**) Cytotoxicity in iRECs at 36 hr treatment with PA 250 μM plus the inhibitors of PERK, ATF6, and IRE1α. Fold representation of object counts/confluence. (**D**) Cytotoxicity in iRECs at 24 hr treatment with PA 250 μM plus the SCD1 inhibitor. Fold representation of object counts/confluence. Data are presented as mean ± SEM. *p<0.05, **p<0.01, ***p<0.001, ****p<0.0001, ns = nonsignificant; two-way ANOVA and Holm–Sidak's multiple comparisons test; (**D**) n = 4; (**A–C, E**) n = 3.

*Figure 3 continued on next page*

*Figure 3 continued*

The online version of this article includes the following source data and figure supplement(s) for figure 3:

**Figure supplement 1.** Mitochondrial fatty acid uptake and autophagy during palmitic acid (PA)-induced endoplasmic reticulum (ER) stress.

**Figure supplement 1—source data 1.** Unedited Western blots of LC3 and p62.

*supplement 1B*). In order to investigate the autophagic flux, we analyzed p62 and LC3 I/II protein levels of cells treated with PA by immunoblotting. While there was no significant change in LC3I/II levels between BSA and BSA-PA-treated cells, p62 was clearly increased in BSA-PA-treated cells. The addition of the lysosomal inhibitor bafilomycin A1 caused an increase in LC3I/II and an additional increase in p62, which was stronger in PA-treated cells vs. control (*Figure 3—figure supplement 1C and D*). This suggests that the accumulation of p62 occurs in the presence of normal autophagic clearance, possibly because the amount of misfolded proteins due to ER stress exceeds the autophagic clearance capacity. However, due to the presence of LC3 puncta and the transcriptional regulation of autophagy in PA-treated cells, it cannot be fully excluded that PA also affects the autophagic process itself.

Altogether, the data demonstrate that ER stress (and p62 accumulation as a consequence thereof) can be induced by PA. Importantly, we also show that the protective effect of OA involves the suppression of ER stress.

## PA increases membrane order in the ER

As previous studies have shown that ER stress can be induced by increased membrane order (*Halbleib et al., 2017*; *Volmer et al., 2013*), we next studied the effect of the fatty acid treatments on the ER membrane order. For this, we made use of C-Laurdan, an anisotropic dye that is able to visualize the degree of membrane order. After staining the cells, we segmented the perinuclear regions and quantified the generalized polarization (GP) values, which are indicative of membrane order. PA treatment decreased ER membrane fluidity (as evidenced by increased GP values) whereas OA increased it. The addition of OA to PA restored the GP values (*Figure 4A and B*) to normal levels. Thus, these results suggest that higher membrane order is associated with the PA-induced ER stress and that the mechanism by which OA suppresses ER stress involves restoring ER membrane order to homeostatic levels.

## PA impairs TAG synthesis and causes the accumulation of disaturated TAG precursors and lysophospholipids

To identify the lipids that might mediate PA-induced changes in ER membrane order, iRECs treated with BSA, BSA-PA, BSA-OA, and BSA-PA/OA were subjected to shotgun lipidomics and quantified. The results are represented schematically in *Figure 5A–D*. All four treatments did not differ much in the relative amount and degree of saturation of the major glycerophosholipid species (phosphatidylcholines, phosphatidylethanolamines, phosphatidylserines, phosphatidylglycerols, and phosphatidylinositols). However, while OA treatment led to the strong formation of TAGs, PA-treated cells presented increased levels of precursors of TAGs synthesis, such as diacylglycerols (DAG), phosphatidic acid (PhA), and lysophosphatidic acid (LPA). The addition of OA to PA enhanced the formation of TAGs, thereby reducing the accumulation of DAG, PhA, and LPA. Moreover, PA treatment dramatically increased the saturation index of DAG, PhA (*Figure 5E and F*), and LPA, and, again, this was fully reverted by OA co-treatment. OA also decreased the concentration of free fatty acids (*Figure 5G*), most likely as a consequence of the stimulated TAG synthesis.

Additionally, we observed a general increase in lysophospholipid levels (lysophosphatidylcholines, lysophosphatidylethanolamines, lysophosphatidylserines, etc.) by PA that was accompanied by the transcriptional down- and upregulation *Lpcat1* and *Lpcat3*, respectively. As part of the Lands' cycle, LPCAT1 prefers palmitoyl-CoA (16:0-acyl-CoA) as an acyl donor to synthesize dipalmitoyl phosphatidylcholine, whereas LPCAT3 favors polyunsaturated FA-CoA as substrates (*Wang and Tontonoz, 2019*). The downregulation of LPCAT1 therefore likely reflects an attempt to dampen the production of oversaturated phosphatidylcholine species and, thereby, the lipid stress in the ER membrane. Interestingly, *Lpcat1* and *Lpcat3* are regulated in the reverse direction in OA-treated cells, demonstrating that a proper level of phosphatidylcholine unsaturation is critical for cell homeostasis (https://doi.org/10.5061/dryad.gqnk98sq7).

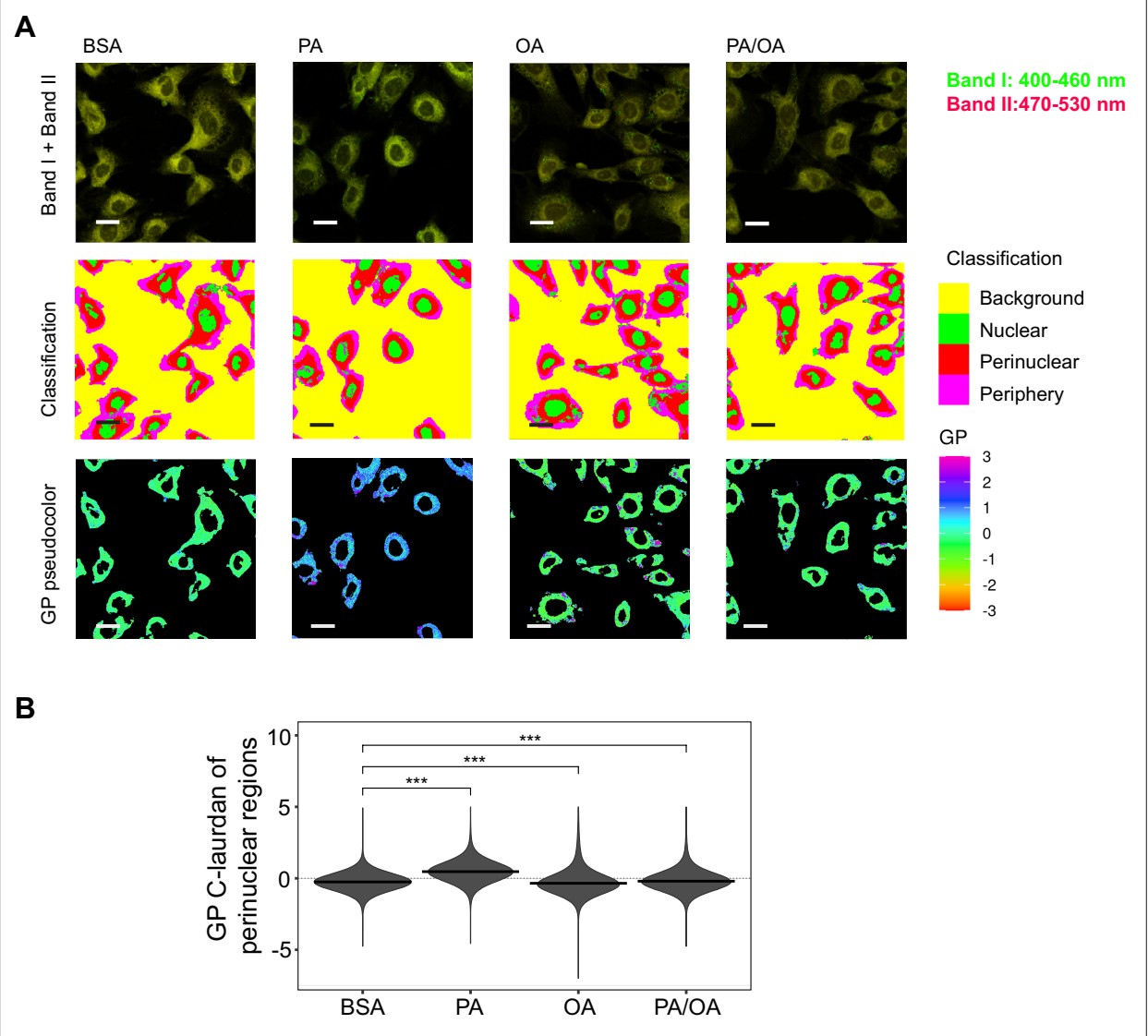

**Figure 4.** Perinuclear membrane order is increased by palmitic acid (PA) treatment and decreased by oleic acid (OA). (**A**) Representative C-Laurdan images of merged channels 1 and 2 (upper row), pixel classification (middle row), and generalized polarization (GP) pseudocolored images (lower row). Induced renal epithelial cells (iRECs) were treated for 16 hr with bovine serum albumin (BSA), PA 250 μM, OA 250 μM, and PA 250 μM plus OA 125 μM. Scale bars: 20 μm. (**B**) GP values quantification of pixels classified as perinuclear from a single representative experiment. *p<0.05, **p<0.01, ***p<0.001, ****p<0.0001, ns = nonsignificant; Wilcoxon signed-rank test; n = 3.

By computing biweight midcorrelation (bicor) (*Song et al., 2012*) between TF activities and lipids abundances, we also identified TF activities that associated with the TAG precursors. We revealed 22 TFs either negatively correlating with monounsaturated PhA and DAG or positively correlating with disaturated PhA and DAG or both (*Figure 5—figure supplement 1A*) as well as 12 TFs associating positively with disaturated and negatively with monounsaturated lysophosphatidylcholine, lysophosphatidylethanolamine, and lysophosphatidylinositol. Both lists include SREBF2 and HNF4A that have established roles in regulating many enzymes important for lipid metabolism (*Figure 5—figure supplement 1B*; *Guan et al., 2011*; *Madison, 2016*; *Wang and Tontonoz, 2019*; *Yin et al., 2011*).

Altogether, our results support previous findings that, unlike MUFAs, SFAs impair TAG production in cultured cells (*Listenberger et al., 2003*; *Piccolis et al., 2019*). The accumulation of LPA, PhA, and DAG further highlights the incapability of DGAT1 and DGAT2 to synthesize TAGs when too many acyl chains are saturated. Moreover, changes in the transcriptome reflect cellular attempts to maintain a proper level of phospholipid unsaturation, most likely to protect against ER stress. Finally, our

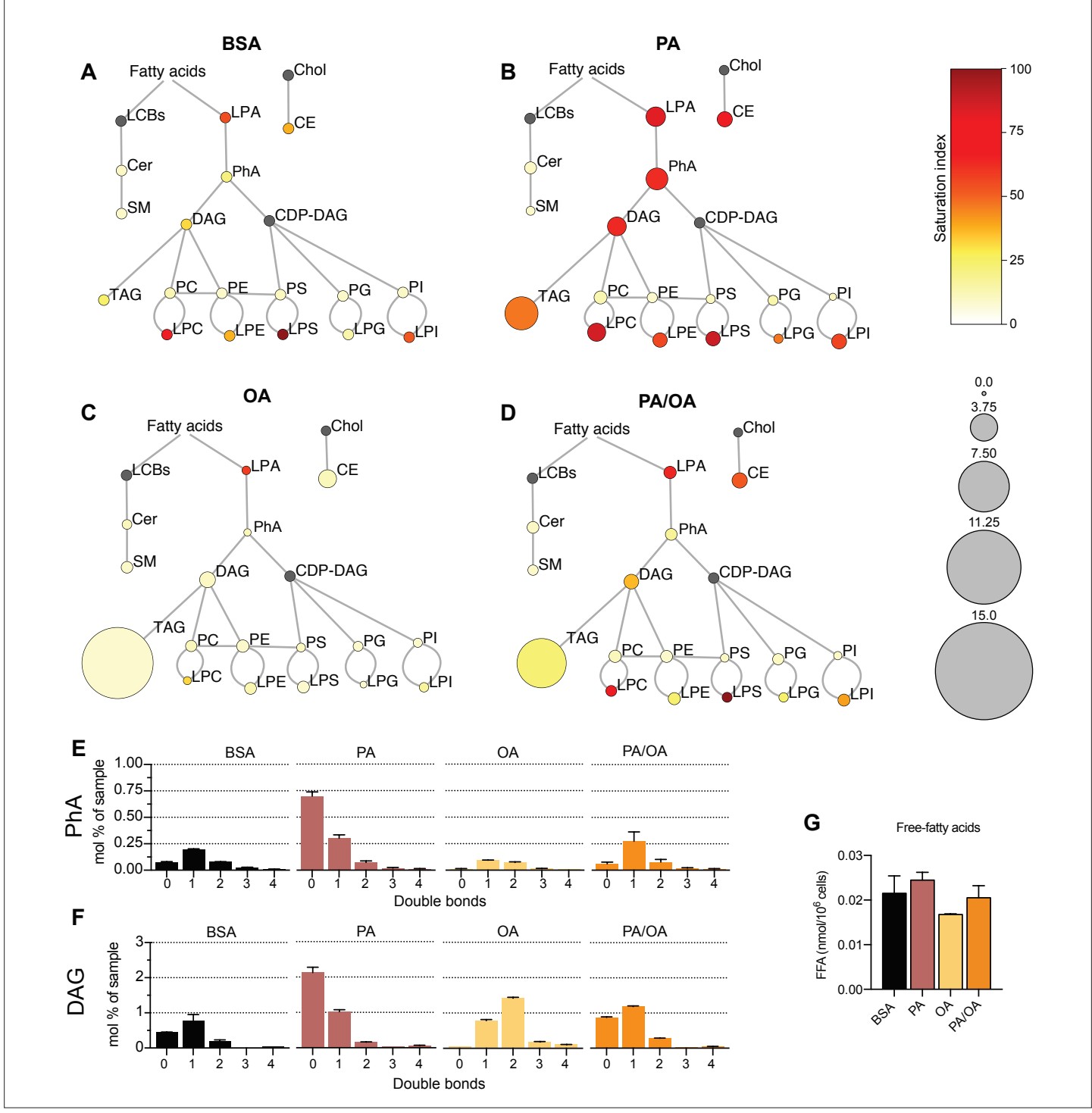

**Figure 5.** Lipidome of induced renal epithelial cells (iRECs) exposed to bovine serum albumin (BSA)-bound fatty acids. (**A–D**) Lipidome of iRECs treated for 16 hr with BSA (**A**), palmitic acid (PA) 250 μM (**B**), oleic acid (OA) 250 μM (**C**), and PA 250 μM plus OA 125 μM (**D**). The scheme shows the relative levels of lipid classes presented as color-coded circles. The lipid species were designated as saturated if all of their fatty acid chains were saturated, or unsaturated if they had at least one unsaturated fatty acid chain. The percentage of saturated lipid species is shown for each class from yellow (low saturation) to red (high saturation). Lipid classes not identified are shown in gray. Cholesterol is also presented in gray because it has no fatty acid chain. The size of the circles is set to the arbitrary unit of 1 for the BSA cells. G3P: glycerol-3-phosphate; LPA: lyso-phosphatidic acids; PhA: phosphatidic acids; DAG: diacylglycerol; TAG: triacylglycerol; PC: phosphatidylcholine: PE: phosphatidylethanolamine; LPE: lyso-phosphatidylethanolamine; LPC: lyso-phosphatidylcholine; PS: phosphatidylserine; LPS: lyso-phosphatidylserine; PI: phosphatidylinositol; LPI: lyso-phosphatidylinositol; PG: phosphatidylglycerol; LPG: lyso-phosphatidylglycerol; Cer: ceramide; SM: sphingomyelin; LCB: long-chain base; CDP: cytidine diphosphate; Chol:

*Figure 5 continued on next page*

Figure 5 continued

cholesterol; CE: cholesterol esthers (n = 3). (**E, F**) Relative amount of PhA (**E**) and DAG (**F**) species classified by the number of double bonds in iRECs treated for 16 hr with BSA, PA 250 μM, OA 250 μM, and PA 250 μM plus OA 125 μM. Data are presented as mean ± SEM; n = 3. (**G**) Cytosolic-free fatty acids in iRECs treated for 16 hr with BSA, PA 250 μM, OA 250 μM, and PA 250 μM plus OA 125 μM. Data are presented as mean ± SEM; one-way ANOVA and Holm–Sidak's multiple comparisons test; n = 3.

The online version of this article includes the following figure supplement(s) for figure 5:

**Figure supplement 1.** Correlation of transcription factor (TF) activity with relevant components of the endoplasmic reticulum (ER) membrane.

multi-omic approach identified TFs potentially sensing ER membrane order and driving adaptation mechanisms.

## LD formation is stimulated by lipid unsaturation

As increased TAG synthesis allows for deposition of lipids in LDs, we next studied whether or not TAG levels correlate with LDs in iRECs. For this, we incubated the cells with BODIPY. OA treatment (0,25 mM) resulted in both higher number and higher size of LDs compared to PA (0.25 mM). Combined treatment (PA 0.25 mM + OA 0.125 mM) resulted in LDs smaller in size compared to OA alone (*Figure 6A–C*). Accordingly, the SCD1 inhibition blocked the formation of LDs in PA-treated cells, resulting in LD numbers and sizes comparable to the BSA group (*Figure 6A, D and E*), suggesting that this enzyme (possibly together with SCD2) is required for the conversion of PA into MUFAs and subsequent LD formation. Moreover, performing a lipid trafficking study with the fatty acid analog BODIPY-C12 showed that, when incubated only with BSA, the BODIPY-C12 dye was located in LDs but also in perinuclear membranes that co-localized with the ER marker calnexin. However, when incubated together with BSA-OA, BODIPY-C12 was strongly directed into LDs (*Figure 6—figure supplement 1A and B*). In sum, MUFAs either imported from extracellular media or produced intracellularly by desaturases enhance the formation of TAGs and, subsequentially, ER-derived LDs.

## LDs protect from PA-induced cytotoxicity

TAG formation is catalyzed by DGAT1 and DGAT2. To test their role in LD formation, we tested inhibitors of DGAT1 (T863) and DGAT2 (PF06424439) at different concentrations. We found that the combination of DGAT1 and DGAT2 inhibitors completely inhibited the formation of LDs (*Figure 7—figure supplement 1A and B*). Moreover, in iRECs treated with BSA or PA/OA, DGAT1/2 inhibitors caused only minimal changes in the relative amount and saturation of each lipid class (*Figure 7A and C*). By contrast, in iRECs treated with PA, the inhibitors caused major disturbances (*Figure 7B*). We observed a massive accumulation of oversaturated TAG precursors (LPA, PhA, and DAG), as well as increased relative levels of saturated lysophospholipids. Especially remarkable is the increase of lysophosphatidylcholine, again reflecting the homeostatic response to prevent the production of oversaturated phosphatidylcholine species. A drastic decrease in cholesterol esters was also observed in all conditions treated with DGAT inhibitors, suggesting that TAG formation influences cholesterol esterification (*Figure 7A–C*).

Using the DGAT inhibitor combination, we could further show that PA-induced cytotoxicity was strongly exacerbated (*Figure 7D*). Accordingly, ROS and ER stress markers were increased upon DGAT inhibition (*Figure 7E and F*). Interestingly, the protective effect of OA concerning cytotoxicity and ER stress was unaltered in the presence of the DGAT inhibitors, suggesting that OA co-treatment can balance the saturation level of ER phospholipids even if TAG formation is blocked. Accordingly, our lipidomics data show that OA co-treatment can reduce disaturated DAG levels (especially DAG 16:0_16:0 and DAG 16:0_18:0) and the ratio between disaturated and monounsaturated DAG compared to PA alone no matter if DGATs are inhibited or not (*Figure 7—figure supplement 2A and B*). With regard to ROS formation, the protective effect of OA was attenuated but did not reach statistical significance. Our results thus suggest that blocking TAG and LD formation, respectively, through DGAT inhibition aggravates PA-induced cell death and ER stress because saturated TAG precursors accumulate in the ER membrane. In this situation, OA treatment can still rescue the lipotoxic effects by influencing DAG composition.

As LDs are dynamic organelles able to fine-tune the release of fatty acids in a lipolysis and re-esterification cycle (*Chitraju et al., 2017*), we hypothesized that MUFAs that are already stored in LDs could be released and help to channel PA towards TAG synthesis. To test this hypothesis, we pretreated the

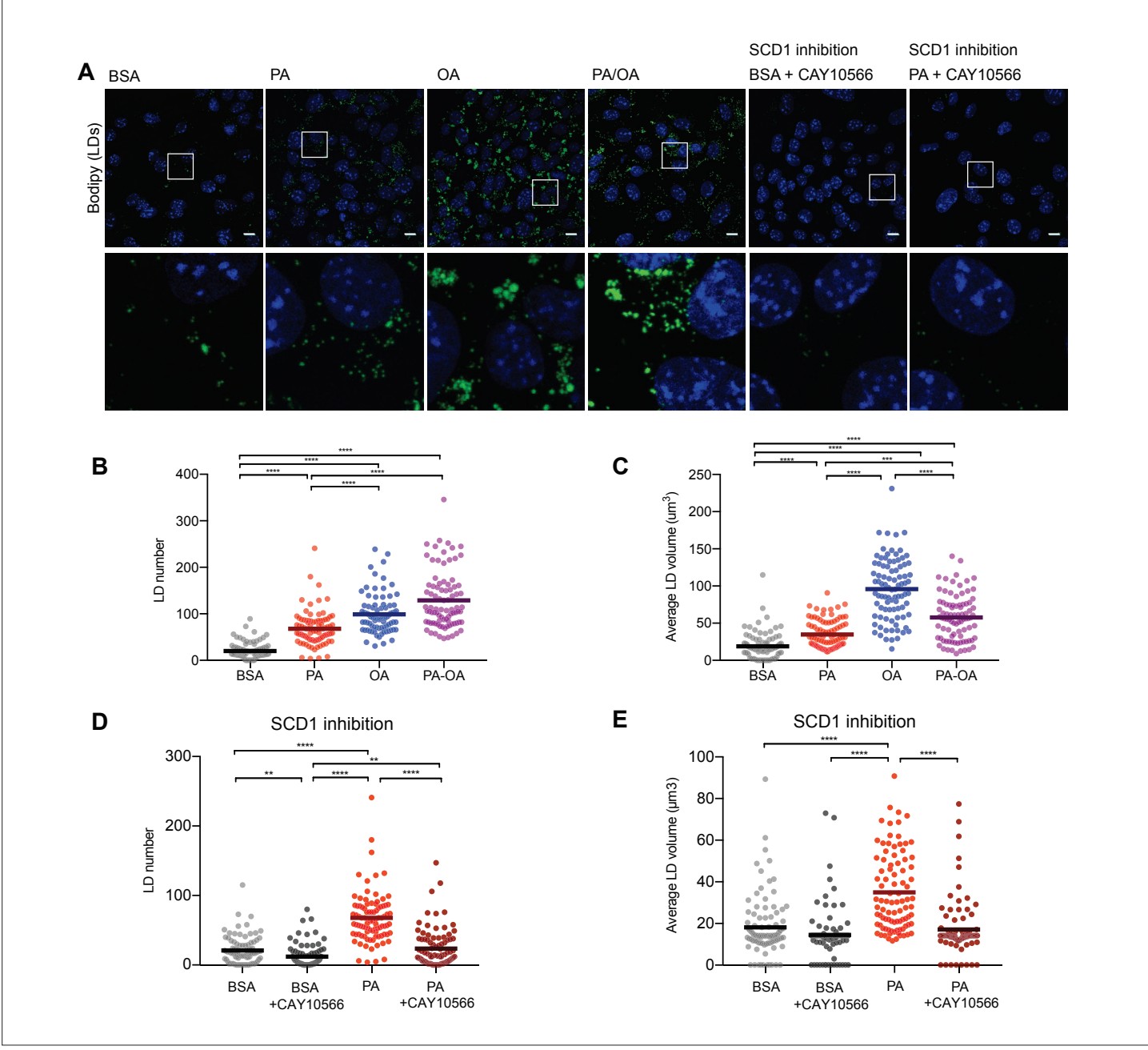

**Figure 6.** Saturated fatty acids impair the formation of lipid droplets (LDs). (**A**) Representative images of LDs stained using BODIPY in induced renal epithelial cells (iRECs) treated for 16 hr with bovine serum albumin (BSA), palmitic acid (PA) 250 µM, oleic acid (OA) 250 µM, PA 250 µM plus OA 125 µM, BSA plus the SCD1 inhibitor CAY10556 (2.5 µM) and PA plus CAY10556 (2.5 µM). Scale bars: 10 µm. (**B–E**) Quantification of LD number (**B, D**) and LD average volume (**C, E**) in iRECs treated for 16 hr with BSA, PA 250 µM, OA 250 µM, PA 250 µM plus OA 125 µM, BSA plus the SCD1 inhibitor CAY10556 (2.5 µM) and PA plus CAY10556 (2.5 µM). Every dot represents the measurement in one single cell. Data information: in (**B–E**), data are presented as the mean + all values. *p<0.05, **p<0.01, ***p<0.001, ****p<0.0001; Kruskal–Wallis plus Dunn's multiple comparisons test. (**B–E**) 10 cells per field from three fields were analyzed for three independent biological replicates.

The online version of this article includes the following figure supplement(s) for figure 6:

**Figure supplement 1.** Oleic acid (OA) channels a fatty acid analog into lipid droplets.

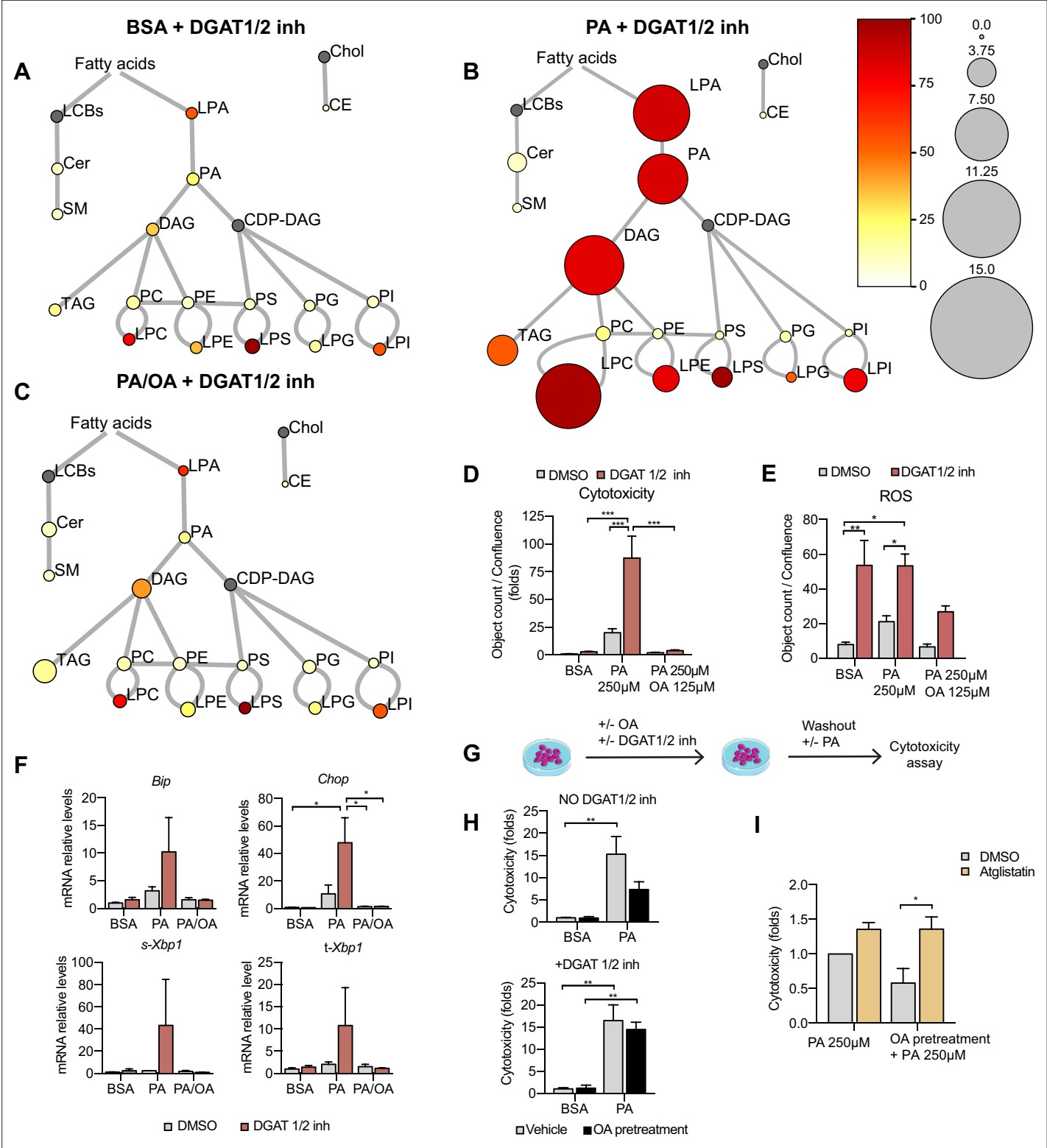

**Figure 7.** Triacylglycerol (TAG) synthesis protects from cytotoxic effects induced by exposure to saturated fatty acids. (**A–C**) Lipidome of induced renal epithelial cells (iRECs) treated for 16 hr with bovine serum albumin (BSA) + T863 30 μM + PF 06424439 30 μM (**A**), palmitic acid (PA) 250 μM + T863 30 μM + PF 06424439 30 μM (**B**), and PA 250 μM + oleic acid (OA) 125 μM + T863 30 μM + PF 06424439 30 μM (**C**). The scheme shows the relative levels of lipid classes presented as color-coded circles. The lipid species were designated as saturated if all of their fatty acid chains were saturated or unsaturated if they had at least one unsaturated fatty acid chain. The percentage of saturated lipid species is shown for each class from yellow (low

*Figure 7 continued on next page*

*Figure 7 continued*

saturation) to red (high saturation). Lipid classes not identified are shown in gray. Cholesterol is also presented in gray because it has no fatty acid chain. The size of the circles is set to the arbitrary unit of 1 for the BSA (*Figure 5*). For lipid classes abbreviations, please refer to the legend of *Figure 5* (n = 3). (**D**) Cytotoxicity in iRECs at 36 hr treatment with BSA, PA 250 µM, and PA 250 µM plus OA 125 µM with or without the DGAT1/DGAT2 inhibitors T863/PF 06424439 (30 µM). Fold representation of object counts/confluence. (**E**) Quantification of reactive oxygen species (ROS) generation in iRECs treated 16 hr with BSA, PA 250 µM, and PA 250 µM plus OA 125 µM with or without the DGAT1/DGAT2 inhibitors T863/PF06424439 (30 µM). Data are presented as object count per well normalized by confluence. (**F**) Quantitative RT-PCR detection of endoplasmic reticulum (ER) stress markers in iRECs treated 16 hr with BSA, PA 250 µM, and PA 250 µM plus OA 125 µM with or without the DGAT1/DGAT2 inhibitors T863/PF06424439 (30 µM). (**G**) Schematic representation of OA pretreatment plus PA insult experiment. (**H**) Cytotoxicity in iRECs at 36 hr after PA 250 µM treatment. Cells were pretreated for 16 hr with OA 500 µM or BSA with or without DGAT1/DGAT2 inhibitors T863/PF06424439 (30 µM). Fold representation of object counts/confluence. (**I**) Cytotoxicity in iRECs at 36 hr after PA 250 µM treatment with or without atglistatin (25 µM). Cells were pretreated for 16 hr with OA 500 µM or BSA. Fold representation of object counts/confluence. In (**D–F, H, I**), data are presented as mean ± SEM. *p<0.05, **p<0.01, ***p<0.001, ****p<0.0001; two-way ANOVA and Holm–Sidak's multiple comparisons test. (**A–D, F, H, I**) n = 3. (**E**) n = 4.

The online version of this article includes the following figure supplement(s) for figure 7:

**Figure supplement 1.** Pharmacological inhibition of lipid droplet (LD) biogenesis and degradation.

**Figure supplement 2.** Oleic acid (OA) rescues palmitic acid (PA)-induced increase of diacylglycerol (DAG) saturation independently of triacylglycerol (TAG) synthesis.

cells with OA 0.5 mM with or without DGAT inhibitors and after washout, we challenged them with PA. Pretreatment with OA reduced PA-mediated cytotoxicity. When TAG synthesis was inhibited through DGAT inhibition, OA pretreatment did not show any effect on PA-induced cytotoxicity, suggesting that LDs rich in OA could protect from PA lipotoxicity (*Figure 7G and H*). To study the underlying mechanism, we used the ATGL inhibitor atglistatin to block the lipolytic release of fatty acids from LDs. Treatment with atglistatin in normal and starved conditions caused an increase in cell area occupied by LDs (*Figure 7—figure supplement 1B and D*). In order to test whether the release of fatty acids from LDs protects against PA-induced cytotoxicity, we pretreated cells with OA and exposed them to PA with and without the ATGL inhibitor. ATGL inhibition caused a significant increase in PA-induced cytotoxicity when compared to untreated cells (*Figure 7I*). Our results, therefore, suggest that LDs serve as a reservoir of MUFAs that can be released via lipolysis to buffer an overload of SFAs. MUFAs released from LDs could then facilitate the incorporation of SFAs into TAGs or decrease the packing density of ER membrane.

## An HFD enriched in SFA increases DAG saturation and induces injury in tubules without LDs

Our cell culture experiments pointed towards disturbances in TAG synthesis as a key mechanism driving lipotoxicity in DKD. In order to validate these findings in our diabetic mouse model, we performed lipidomic studies on the kidney cortex of STZ-injected mice fed a chow diet, MUFA-HFD, and SFA-HFD. In agreement with the histological LD profile of the kidney cortex in these different conditions, the diet enriched in MUFA stimulated the synthesis of TAGs while the diet enriched in SFA increased the saturation levels of the DAG species and impaired TAG synthesis (*Figure 8A*). Thus, the chronic dietary interventions in the mice have strikingly similar effects on the lipidome as the more acute cell culture treatments.

Finally, we asked whether the increased TAG and LD levels spatially correlate with the enhanced kidney injury in the STZ + HFD conditions. Since renal lipid accumulation in our diabetic mouse model was not uniform, we examined the distribution of KIM-1 and LDs within the cortical kidney tissue. Interestingly, in both STZ-HFD groups, the vast majority of the LD-positive tubules were not marked by the KIM-1 antibody. As reported above, in the SFA-HFD mice there were more tubules than in MUFA-HFD mice showing increased KIM-1 expression and these were mostly devoid of LDs. Vice versa, in the MUFA-HFD mice there were more LD-positive cells that in turn were mostly KIM-1-negative (*Figure 8B and C*). Hence, it can be concluded that tubular damage is mainly occurring in the LD-poor tubules.

In order to identify the tubular segments positive for KIM-1 in SFA-HFD mice, we investigated the spatial distribution of KIM-1, the S1 marker SGLT2, and the S1-S3 marker megalin. KIM-1 immunoreactivity was seen in cortical PTCs positive for megalin (*Figure 8—figure supplement 1A*) and SGLT2 (*Figure 8—figure supplement 1B*), while the corticomedullary junction was mostly devoid of KIM-1.

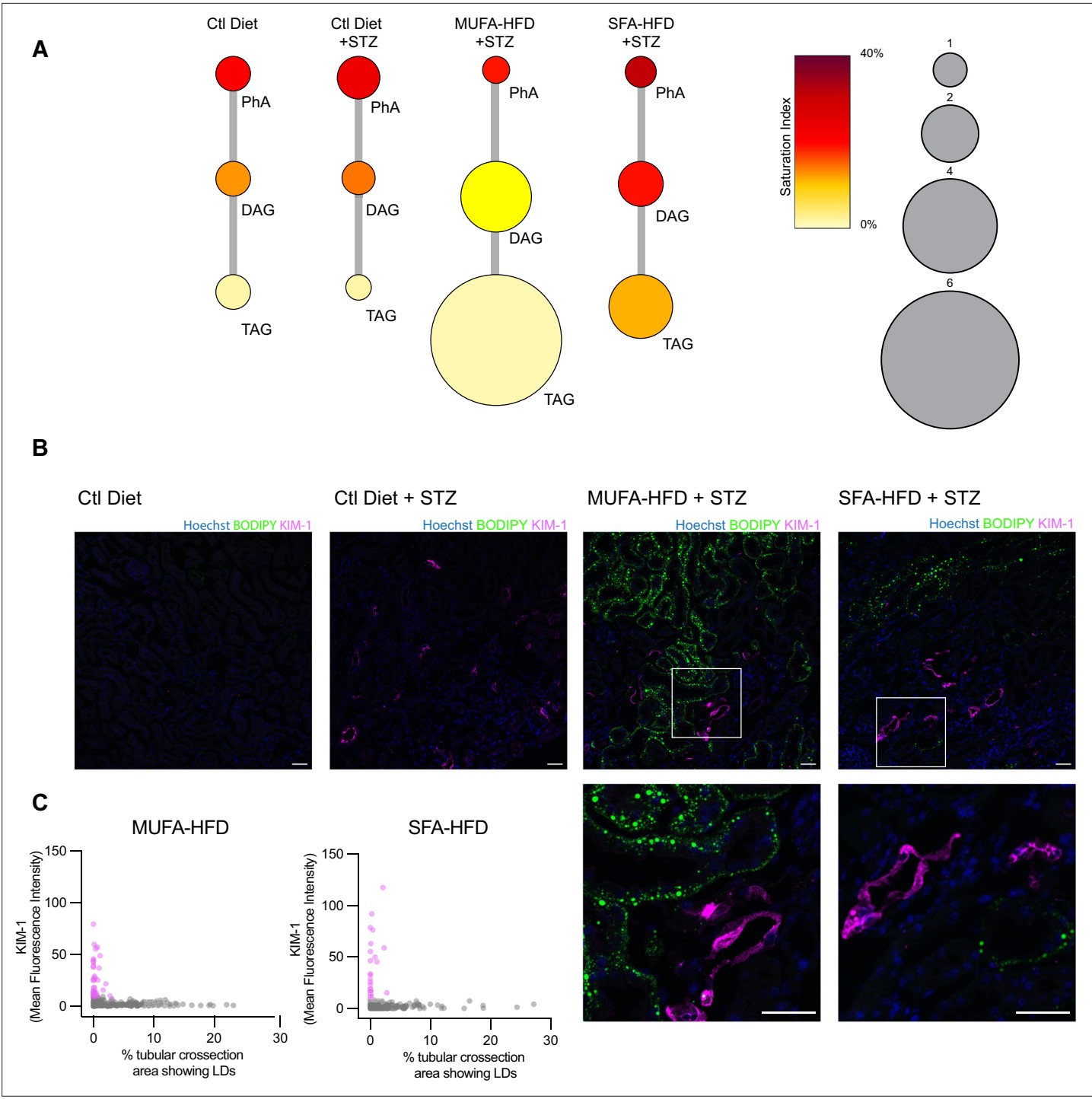

**Figure 8.** Triacylglycerol (TAG) synthesis protects from cytotoxic effects induced by exposure to saturated fatty acids in mouse kidney. (**A**) Relative levels of lipid classes involved in the synthesis of TAGs in the kidney cortex of mice. Lipid species were designated as saturated if both of their acyl chains were saturated or unsaturated if they had at least one unsaturated acyl chain. The size of the circles is set to the arbitrary unit of 1 for the control group. PhA: phosphatidic acid; DAG: diacylglycerol; TAG: triacylglycerol. n = 3. (**B, C**) Representative confocal fluorescence images (**B**) and quantification (**C**) of mouse kidney stained for KIM-1 and lipid droplets. Scale bars: 50 µm. (**C**) Each dot represents one tubular cross section. All megalin-positive tubules from three images from two mice were analyzed for each condition. Dots are colored magenta when KIM-1 mean intensity was bigger than 10.

The online version of this article includes the following figure supplement(s) for figure 8:

**Figure supplement 1.** KIM-1 colocalizes with SGLT2 and megalin in the cortex of diabetic mice.

**Figure supplement 2.** KIM-1 does not colocalize with SGLT2 and megalin in a subset of tubules in the cortex of diabetic mice.

KIM-1 expression was also observed in severely damaged tubules that showed partial loss of the segment markers (*Figure 8—figure supplement 2*). Altogether, these results support the protective role of TAG/LD formation and provide in vivo validation of the mechanisms leading to lipotoxicity in diabetes.

## Discussion

Lipids and lipid metabolites accumulate in tubules from humans and animal models of DKD, suggesting that lipotoxicity contributes to DKD pathogenesis (*Herman-Edelstein et al., 2014*; *Kang et al., 2015*). While the clinical practice recommendation for DKD by the KDIGO 2020 (*Kidney Disease: Improving Global Outcomes (KDIGO) Diabetes Work Group, 2020*) initiative (Kidney Disease: Improving Global Outcomes Diabetes Work Group, 2020) generally suggested the preferred intake of unsaturated over saturated fat, more emphasis was placed on low protein and low sodium diets. So far only limited data is available on the importance of monounsaturated fat in the dietary management of DKD in humans. While one study showed no protective effect of a Mediterranean diet compared to a low-fat control diet with regard to the DKD incidence in type 2 diabetics (*Diaz-Lopez et al., 2015*), two studies demonstrated renoprotective effects with adherence to Dietary Approaches to Stop Hypertension (DASH) and Mediterranean diets in cohorts of diabetic women (*Jayedi et al., 2019*; *Yu et al., 2012*).

In our mouse model, in which STZ-induced hyperglycemia was combined with an HFD, we found a depletion of the white adipose tissue that was accompanied by a significant weight gain in liver and kidney. Accordingly, the circulating lipids were increased, and in the kidney, lipid deposition was found in the tubular cells of the cortex. This suggests that, in addition to increased dietary fat intake, lipolysis in white adipose tissue might have contributed to renal fat accumulation. Strikingly, lipid deposition was primarily observed in the straight S2/S3 PTC segments while omitting glomeruli, S1 PTC segments, and distal tubules. This lipid pattern could be explained by differences in fat uptake between the different nephron segments. As the activity of the main albumin receptor cubilin is known to be restricted to the S1 segment (*Christensen et al., 2021*; *Ren et al., 2020*), other uptake pathways may account for lipid uptake in S2/S3. Uptake pathways that we tested are the apical FATP2 and the basolateral CD36 fatty acid transporters (*Khan et al., 2018*; *Susztak et al., 2005*). As seen in *Figure 1—figure supplement 2C*, our expression analysis showed strong downregulation of both genes in all three mice groups treated with STZ compared to the control group, which may argue against an important role for them in DKD. Alternatively, it was recently shown that the scavenger receptor KIM-1 mediates apical albumin-PA uptake in PTCs to promote DKD (*Mori et al., 2021*). However, as we detected an inverse correlation between KIM-1 expression and LDs, it seems at least unlikely that KIM-1 can promote lipid storage. Therefore, it is more conceivable that the preferential lipid accumulation in S2/S3 is caused by differences in lipid metabolism. Indeed, it is known that PTCs in the straight segments contain less mitochondria (*Hall et al., 2021*; *Maunsbach, 1966*), possibly resulting in LD build-up due to decreased β-oxidation. Also, restricted expression of SCD1 to the straight S2/S3 segments could explain the lipid deposition pattern (*Zhang et al., 2006*).

Upon cell entry, the intracellular fate and toxicity of the fatty acids clearly depended on the saturation of the acyl chain. While lipid storage in LDs was stronger in the MUFA-HFD-treated mice than in SFA-HFD-treated mice, tubular damage was reduced. This argues for a protective role of the MUFA diet in the tubules, which was corroborated by our findings that the cells showing KIM-1 expression were mostly devoid of LDs. However, as our mouse experiment was ended already 20 weeks after introducing the HFD, it would be interesting to study more long-term effects on renal function.

Using our cell culture model, we identified both mitochondrial and ER homeostasis as the main determinants of cell viability during PA-induced lipotoxic stress. The functional link between the two organelles was revealed by the finding that the inhibition of mitochondrial fatty acid uptake by etomoxir worsened ER stress and cytotoxicity induced by PA. Free SFAs that cannot be oxidized in mitochondria were found to be incorporated into TAG precursors causing ER lipid bilayer stress. This could be suppressed by the addition of OA that promoted TAG and LD formation, which was also reflected by a reduction of free fatty acids by OA.

Due to the high degree of unsaturated lipids, the ER membrane is normally one of the most fluid membranes in the cell (*Barelli and Antonny, 2016*). Using Laurdan imaging, the ER membrane showed higher packing density upon PA treatment, most likely due to the accumulation of disaturated TAG precursors, in particular DAGs. This is in agreement with previous findings that disaturated DAGs

are a poor substrate for DGAT activity and that the inhibition of GPAT enzymes that catalyze the first addition of fatty acids to the glycerol backbone is a promising approach for preventing lipotoxic cell injury (*Piccolis et al., 2019*). For diabetic kidneys, this might be particularly important as DAGs have been implicated in insulin resistance due to their role in activating protein kinase C (PKC) isoforms (*Lyu et al., 2020*). How disaturated DAGs cause ER stress is not fully understood, but studies have shown that this likely involves PERK and IRE1-induced sensing of membrane lipid saturation (*Halbleib et al., 2017*; *Volmer et al., 2013*). Additionally, the altered ER membrane environment may lead to the misfolding of transmembrane proteins and subsequent induction of the unfolded protein response (UPR). Finally, the impairment of LD formation by PA may lower the capacity to buffer the lipotoxic stress in the ER.

The main finding of our study is that all observed PA-induced cytotoxic effects could be suppressed by adding OA. OA increased DGAT-mediated TAG, which in turn facilitated LD formation. Both in cell culture and in the mouse kidneys, LD formation correlates positively with the protective effects of MUFAs and negatively with cell damage. Yet, LD formation was also shown to be dispensable for OA-mediated rescue effects in PA-induced stress because OA can reduce disaturated DAGs even in the presence of DGAT inhibitors. Only when cells were pretreated with OA, then OA-mediated rescue effects were DGAT-dependent, suggesting that the preexistence of LDs is relevant for these rescue effects and/or that the beneficial effects of LD formation lag behind those associated with the formation of unsaturated phospholipids. As we also show that ATGL-mediated lipolysis is required for the OA rescue effect, our interpretation is that LDs can function as a reservoir for unsaturated lipids that can be released, for example, when the ER membrane desaturation is increased. How this crosstalk between the ER and LDs could be regulated is an interesting question and should be subject of further studies.

In summary, we have undertaken a comprehensive analysis of PTC responses to lipotoxic stress. We find that mice exposed to HFD show a striking lipid accumulation pattern with more LDs in the straight S2/S3 PTCs. As these segments also seem to be the site where less tubular injury occurs, this points towards a protective role of lipid storage. Accordingly, MUFA-HFD that leads to more LDs is associated with less injury while the opposite is true for SFA-HFD. Mechanistically, we identify ER membrane saturation as a key determinant of cell viability that is regulated by LDs as a reservoir for MUFAs. Moreover, we identify transcriptional networks activated during PA-induced stress that can be used as a resource for a systems-level understanding of lipotoxic stress. As dietary effects on DKD progression are so far understudied in humans, our findings provide new rationales for emphasizing MUFAs in the dietary management of DKD.

# Materials and methods

**Key resources table**

| Reagent type (species) or resource | Designation | Source or reference | Identifiers | Additional information |
|---|---|---|---|---|
| Commercial assay or kit | Free fatty acid quantification Kit | Sigma-Aldrich | #MAK044 | |
| Commercial assay or kit | Triglyceride determination kit | Sigma-Aldrich | #MAK266 | |
| Commercial assay or kit | Mouse NGAL (Lipocalin-2) ELISA Kit | BioLegend | #443,707 | |
| Chemical compound, drug | T863 | MedChemExpress | HY-32219 | (30 µM) |
| Chemical compound, drug | PF 06424439 | Bio-Techne | #6348/5 | (30 µM) |
| Chemical compound, drug | Etomoxir | Calbiochem | #236,020 | (50 nM; 200 nM) |
| Chemical compound, drug | Bafilomycin A1 | VWR | #J61835.MX | (200 nM) |
| Chemical compound, drug | CAY10566 | MedChemExpress | #HY-15823 | (2.5 µM; 5 µM) |
| Chemical compound, drug | Atglistatin | Sigma-Aldrich | SML1075 | (25 µM) |
| Chemical compound, drug | GSK2606414 | MedChemExpress | HY-18072 | (5 µM; 10 µM) |
| Chemical compound, drug | GSK2850163 | Sigma-Aldrich | #1,684 | (10 µM; 30 µM) |

*Continued on next page*

*Continued*

| Reagent type (species) or resource | Designation | Source or reference | Identifiers | Additional information |
|---|---|---|---|---|
| Chemical compound, drug | Ceapin-A7 | Sigma-Aldrich | #SML2330 | (1 µM; 2 µM) |
| Chemical compound, drug | Tunicamycin | Sigma-Aldrich | #5045700001 | (10 µM) |
| Chemical compound, drug | Thapsigargin | Sigma-Aldrich | #586,005 | (1 µM) |
| Chemical compound, drug | BODIPY 493/503 | Thermo Fisher Scientific | #D3922 | (2.5 µg/mL) |
| Chemical compound, drug | C1-BODIPY 500/510C12 | Thermo Fisher Scientific | #D3823 | (2 µM) |
| Chemical compound, drug | C-Laurdan | Dr. B.R. Cho, Korea University, South Korea | PMID:23311388 | (10 µg/mL) |
| Chemical compound, drug | IncuCyte Cytotox Red Reagent | EssenBio | #4632 | (1:2000) |
| Chemical compound, drug | TMRE | Thermo Fisher Scientific | #T669 | (50 nM) |
| Chemical compound, drug | DHE | Thermo Fisher Scientific | #D23107 | 10 µM |
| Chemical compound, drug | Streptozotocin; STZ | Sigma-Aldrich | #S0130 | (50 mg/kg) |
| Antibody | Anti-p62 (rabbit, polyclonal) | Cell Signaling | #5114 | WB (1:1000) |
| Antibody | Anti-LC3 (rabbit, polyclonal) | MBL | #PM036 | WB (1:1000) |
| Antibody | Anti-GAPDH (rabbit, polyclonal) | Abcam | #ab9485 | WB (1:1000) |
| Antibody | G anti-KIM-1 (goat, polyclonal) | R&D Systems | #AF1817 | IHC (1:100) |
| Antibody | Anti-SGLT2 (rabbit, polyclonal) | Abcam | #ab85626 | IHC (1:200) |
| Antibody | Anti-p62 (guinea pig, polyclonal) | Progen | #GP62-C | IF (1:1000) |
| Antibody | Anti-megalin (rabbit, polyclonal) | Prof. Michigami, Osaka Women's and Children's Hospital, Osaka, Japan | PMID:15976002 | IHC (1:1000) |
| Cell line (*Mus musculus*) | iRECs | Prof. Soeren S. Lienkamp. Institute of Anatomy, University of Zurich, Zurich, Switzerland | PMID:27820600 | |
| Cell line (*Homo sapiens*) | HK-2 | ATCC | CRL-2190 | |
| Cell line (*Didelphis marsupialis virginiana*) | OK | ATCC | CRL-1840 | |
| Sequence-based reagent | 18S Fw | This paper | qPCR primers | 5'-CGGCTACCACATCCAAGGAA-3' |
| Sequence-based reagent | 18S Rv | This paper | qPCR primers | 5'-GCTGGAATTACCGCGGCT-3' |
| Sequence-based reagent | *Actb* Fw | This paper | qPCR primers | 5'GCTCTGGCTCCTAGCACCAT-3' |
| Sequence-based reagent | *Actb* Rv | This paper | qPCR primers | 5'-GCCACCGATCCACACAGAGT-3' |
| Sequence-based reagent | *Bip* Fw | This paper | qPCR primers | 5'-TTCAGCCAATTATCAGCAAACTCT-3' |
| Sequence-based reagent | *Bip* Rv | This paper | qPCR primers | 5'-TTTTCTGATGTATCCTCTTCACCAGT-3' |
| Sequence-based reagent | *Cd36* Fw | This paper | qPCR primers | 5'-GATGACGTGGCAAAGAACAG-3' |
| Sequence-based reagent | *Cd36* Rv | This paper | qPCR primers | 5'-TCCTCGGGGTCCTGAGTTAT-3' |
| Sequence-based reagent | *Chop* Fw | This paper | qPCR primers | 5'-CCACCACACCTGAAAGCAGAA-3' |
| Sequence-based reagent | *Chop* Rv | This paper | qPCR primers | 5'-AGGTGAAAGGCAGGGACTCA-3' |
| Sequence-based reagent | *Ccl5* Fw | This paper | qPCR primers | 5'-CCCTCACCATCATCCTCACT-3' |
| Sequence-based reagent | *Ccl5* Rv | This paper | qPCR primers | 5'-TCCTTCGAGTGACAAACACG-3' |
| Sequence-based reagent | *Fn1* Fw | This paper | qPCR primers | 5'-TTAAGCTCACATGCCAGTGC-3' |

*Continued on next page*

*Continued*

| Reagent type (species) or resource | Designation | Source or reference | Identifiers | Additional information |
|---|---|---|---|---|
| Sequence-based reagent | *Fn1* Rv | This paper | qPCR primers | 5'-TTAAGCTCACATGCCAGTGC-3' |
| Sequence-based reagent | *Slc27a2* Fw | This paper | qPCR primers | Fw 5'-ACACACCGCAGAAACCAAATGACC-3' |
| Sequence-based reagent | *Slc27a2* Rv | This paper | qPCR primers | 5'-TGCCTTCAGTGGATGCGTAGAACT-3' |
| Sequence-based reagent | *s-Xbp1* Fw | This paper | qPCR primers | 5'-CTGAGTCCGAATCAGGTGCAG-3' |
| Sequence-based reagent | *s-Xbp1* Rv | This paper | qPCR primers | *5'-GTCCATGGGAAGATGTTCTGG-3'* |
| Sequence-based reagent | *t-Xbp1* Fw | This paper | qPCR primers | 5'-TGGCCGGGTCTGCTGAGTCCG-3' |
| Sequence-based reagent | *t-Xbp1* Rv | This paper | qPCR primers | 5'-GTCCATGGGAAGATGTTCTGG-3' |

## Animal experimentation and diets

All of the experimental protocols were performed with the approval of the animal experimentation ethics committee of the University Paris Descartes (CEEA 34), projects registered as 17-058 and 20-022. Mice were kept in a temperature-controlled room (22°C ± 1°C) on a 12/12 hr light/dark cycle and were provided free access to commercial rodent chow and tap water prior to the experiments. The MUFA-HFD and SFA-HFD were obtained from Research Diets. The HFDs (Research Diets, #D20072102 and #D20072103) had the following composition (in percentage of calories): 20% protein, 35% carbohydrates, and 45% fat. The control diet (ENVIGO, #2018) composition was 24% protein, 58% carbohydrates, and 18% fat. MUFA-HFD fat source was olive oil (95.1%) plus soybean oil (4.90%), and SFA-HFD fat source was butter (95.1%) plus soybean oil (4.90%). HFDs were supplemented with soybean oil to cover the essential need for polyunsaturated fatty acids (PUFAs). The detailed composition is shown in *Appendix 1—table 1*.

C57BL/6NCrl male 7-week-old mice were put on a control diet, MUFA-HFD, or SFA-HFD with free access to food and water. At 11 weeks old, insulin deficiency was induced by intraperitoneal administration of STZ (50 mg/kg per day for five consecutive days). Mice were fasted for 6 hr before STZ injections. STZ (Sigma-Aldrich, #S0130) was freshly prepared in 50 mM sodium citrate buffer pH 4.5 before administration. Control mice were injected with sodium citrate buffer. Blood samples were taken from the mandibular vein before STZ injections and every 4 weeks. Spot urine was collected at 10 and 14 weeks after STZ injection. Metabolic cages were avoided because they could have produced body weight loss and compromised the experiment. Food and water intake were measured per cage and values were divided by the number of mice in each cage.

The animals were sacrificed by cervical dislocation 16 weeks after STZ treatment. Blood was taken by intracardiac puncture, and organs were perfused from the heart with PBS. The tissues (kidney, liver, heart, eWAT) were extracted, weighed, and processed for histology and molecular analysis.

## Plasma and urine parameters

Urinary albumin and creatinine were determined by mouse albumin ELISA quantification kit (Bethyl Laboratories, #E99-134) and a home assay based on the Creatinine Parameter Assay Kit (R&D Systems, #KGE005). Urine glucose levels were measured with a COBAS 2000 analyzer (Roche). Blood glucose levels were measured using an 'On Call GK dual' glucometer (Robe Medical, #GLU114). Blood levels of TAGs were measured by a colorimetric assay using the Triglyceride determination kit (Sigma-Aldrich, #MAK266). LCN2 plasma levels were determined by ELISA (R&D Systems, #AF1857-SP).

## Renal histopathology

Picro-Sirius Red and PAS stainings were performed on paraffin-embedded sections. ORO staining was performed on OCT frozen sections. Images were acquired in a slide scanner Nanozoomer HT2.0 C9600 (Hamatsu). Picro-Sirius Red images were analyzed using ImageJ. Images were converted into RGB stack and the green channel was selected. The cortex region was segmented and the threshold was manually adjusted to determine the percentage of fibrotic area. ORO staining was quantified using the pixel classification tool from QuPath.

## Cell culture

All cell lines were maintained at 37°C and 5% $CO_2$. iRECs were cultured on 0.1% gelatin-coated flasks in Dulbecco's modified Eagle's medium (DMEM) (Lonza, #BE12-604F/U1) supplemented with penicillin/streptomycin (Sigma-Aldrich, #P4333), L-glutamine (Thermo Fisher Scientific, #25030024), and 10% (v/v) fetal bovine serum (FBS) (Thermo Fisher Scientific, #10270106). HK-2 cells were cultured in Renal Epithelial Cell Growth Medium (PromoCell, #C-26030) supplemented with SupplementMix (PromoCell, #C-39606) for a final concentration of fetal calf serum 0.5% (v/v), FBS 1.5% (v/v), epidermal growth factor (10 ng/mL), human recombinant insulin 5 µg/mL, epinephrine 0,5 µg/mL, hydrocortisone 36 ng/mL, human recombinant transferrin 5 g/mL, and triiodo-L-thyronine 4 pg/mL. Primary mouse proximal tubule epithelial cells were isolated from mouse renal cortices as previously described (*Legouis et al., 2015*) and were cultured in Basal Medium 2 phenol red-free (PromoCell, #C-22216) supplemented in the same way as HK-2 cells. OK cells were cultured in DMEM/F12 (Thermo Fisher Scientific, #21331020) supplemented with penicillin/streptomycin, glutamine, and 10% (v/v) FBS. With regard to identity, iRECs were provided directly by their creators (PMID: 27820600). The negative mycoplasma contamination status was confirmed by regular testing.

## BSA fatty acid conjugation

Fatty acid-free BSA (Sigma-Aldrich, #A8806) was added to complete medium to a final concentration of 1% (w/v), PA (Sigma-Aldrich, #P0500), OA (Sigma-Aldrich #O1008), or a combination of both were added to the medium and incubated at 37°C for 30 min.

## Pharmacological treatments

TAG formation was inhibited using the DGAT1 inhibitor T863 (MedChemExpress, #HY-32219) and the DGAT2 inhibitor PF 06424439 (Bio-Techne, #6348/5). Import of fatty acids into mitochondria was blocked by CPT1 inhibitor etomoxir (Calbiochem, #236020). Desaturation of fatty acids was inhibited by the SCD1 inhibitor CAY10566 (MedChemExpress, #HY-15823-1mg). Lipolysis was inhibited by treating the cells with the ATGL inhibitor atglistatin (Sigma-Aldrich, #SML1075). The ER stress response branches were inhibited individually using the PERK inhibitor GSK2606414 (MedChemExpress, #HY-18072), the IRE1a inhibitor GSK2850163 (Sigma-Aldrich, #1684), and the ATF6 inhibitor Ceapin-A7 (Sigma-Aldrich, #SML2330). ER stress was chemically induced with tunicamycin (Sigma-Aldrich, #5045700001) and thapsigargin (Sigma-Aldrich, #586005). Autophagosome-lysosome fusion was blocked using the V-ATPase inhibitor bafilomycin A1 (VWR, #J61835.MX). The concentration of each treatment is indicated in the figure legends.

## Immunoblotting

Whole-cell lysates from IRECs were isolated using RIPA buffer. The extracts were resolved by SDS-polyacrylamide gel electrophoresis and transferred onto an Amersham Protran Premium Nitrocellulose 0.45 µm membrane (Amersham, #10600003). Membranes were blocked for 1 hr at room temperature in PBS 5% milk. The blots were then incubated with primary antibody diluted 1:1000 in PBS 0.1% Tween 1% milk overnight at 4°C. The antibodies used were LC3 (MBL, #PM036), P62 (Cell Signaling, #5114), and GAPDH (Abcam, #ab9485). The blots were washed three times and incubated with horseradish peroxidase-conjugated secondary antibody in PBS 0.1% Tween 1% milk for 2 hr at room temperature. After three washes, the blots were developed using the SuperSignal West Dura Extended Duration Substrate (Thermo Fisher Scientific, #34076). The quantification was done by densitometry using ImageJ software.

## RNA-seq

Total RNA was isolated using the RNeasy Kit (QIAGEN, #74104) including a DNAse treatment step. RNA quality was assessed by capillary electrophoresis using High Sensitivity RNA reagents with the Fragment Analyzer (Agilent Technologies), and the RNA concentration was measured by spectrophotometry using the Xpose (Trinean) and Fragment Analyzer capillary electrophoresis.

RNA-seq libraries were prepared starting from 1 µg of total RNA using the Universal Plus mRNA-Seq kit (Nugen) as recommended by the manufacturer. The oriented cDNAs produced from the poly-A+ fraction were sequenced on a NovaSeq6000 from Illumina (paired-end reads 100 bases + 100 bases). A total of ~50 million of passing filter paired-end reads was produced per library.

## Transcriptomics data processing and analysis

Galaxy platform was used to analyze the transcriptome data (*Afgan et al., 2016*). Quality check was assessed with FastQC v0.11.8 (http://www.bioinformatics.babraham.ac.uk/projects/fastqc/). After trimming with Trim Galore! v0.4.3 (http://www.bioinformatics.babraham.ac.uk/projects/trim_galore/), reads were aligned to the genome assembly GRCm38 using RNA STAR2 v2.5.2b (*Dobin et al., 2013*). Gene counts were calculated with featureCounts v1.6.4 (*Liao et al., 2014*), and differentially expressed gene (DEG) analysis was performed with DESeq2 v1.22.1 (*Love et al., 2014*). Gene counts were turned into log2 scale and used to perform principal components analysis. Volcano plot for differences between PA and BSA treatments was drawn using EnchancedVolcano (https://github.com/kevinblighe/EnhancedVolcano). Significant DEGs were then divided into 12 clusters using the soft clustering tool Mfuzz v2.40.0 (*Kumar and E Futschik, 2007*), and computed clusters were assigned to expected expression change patterns. We investigated more closely the genes with expression patterns changed between BSA and PA treatment as well as PA and PA/OA treatment. Using Bioconductor package clusterProfiler v3.8.1 (*Yu et al., 2012*), enriched GO categories for biological processes were compared and 25 most significant terms were plotted. All the analysis and data visualization were conducted in R v4.1.0.

For the transcription factor activity estimation, T-values of the differential analysis of PA vs. Ctrl, OA vs. Ctrl, OA vs. PA/OA, PA/OA vs. Ctrl, and PA vs. PA/OA were used as input statistic for the VIPER algorithm (*Alvarez et al., 2016*). Regulon of TF and targets were obtained from the dorothea R package (*Holland et al., 2020*). Only TF-target interactions that belong in the confidence classes A, B, and C were kept. The viper function was used with default parameters, except minimum regulon size of 5, eset.filter parameter set to FALSE, and pleiotropy parameter set to FALSE.

Biweight midcorrelation (bicor) was systematically estimated between TFs and lipid abundances using the bicor function of the WGCNA R package. To do so, average TF activities were first estimated at the level of individual conditions using the same method as described in the previously. Then, average TF activities across conditions (Ctrl, OA, PA/OA, PA) were correlated with average lipid abundance across the same conditions. To focus on the top associations between TFs and lipids, we selected TFs-lipids bicor coefficients above 0.97 or under –0.97. This coefficient value was chosen because it allowed us to select a number of associations that could be humanly investigated (i.e., less than 30). The full code for the analysis can be found at https://github.com/saezlab/Albert_perez_RNA_lipid/tree/main/scripts (*Mansouri and Dugourd, 2022*; copy archived at swh:1:rev:e445d92c5197dfc8afca02e47b680d89709efb7b).

## Immunofluorescence and lipid droplet stainings

For immunocytochemistry, cells were washed three times in PBS, fixed for 20 min in 4% paraformaldehyde in PBS, blocked for 10 min in PBS + 3% BSA + 0.1% Tween + 0.1% Triton, and incubated overnight at 4°C with primary antibodies diluted in PBS + 3% BSA + 0.1% Tween + 0.1% Triton. After washing, cells were incubated 2 hr at room temperature with secondary antibodies (dilution 1:1000) and Hoechst (0.5 µg/mL) diluted in PBS + 3% BSA + 0.1% Tween + 0.1% Triton. Cells were washed three times in PBS and mounted in Roti-Mount FluorCare (Roth, HP19.1). As primary antibodies, guinea pig anti-p62 (1:1000, Progen, #GP62-C) and rabbit anti-LC3 (1:500, MBL International, #PM036) were used, and as secondary antibodies fluorescent conjugated Alexa Fluor 647 (Thermo Fisher Scientific, #A21244 and #A21450).

For LD imaging, BODIPY 493/503 (Thermo Fisher Scientific, #D3922) was incubated at 2.5 µg/mL together with secondary antibodies. For lipid trafficking studies, the fatty acid analog C1-BODIPY 500/510 C12 was incubated for 6 hr at a final concentration of 2 µM (Thermo Fisher Scientific, #D3823). Images were acquired on a Leica TCS SP8 equipped with a 405 nm laser line and a white light laser with a ×63/1.4 DIC Lambda blue Plan Apochrome objective. Percentage of total cell area occupied by LDs was quantified with the "Analyze Particles"-tool of Fiji on thresholded Z stack projections images of BODIPY 493/503. LD size and number were measured with Imaris by defining the volume of the LDs in 3D and then using the Spot Detector tool.

## Immunohistochemistry

For immunohistochemistry, kidney sections (4 µm) were generated from paraffin-embedded tissue using a Leica RM2145 microtome and mounted on glass slides. Sections were first deparaffinized with

xylene and rehydrated using a graded ethanol series. Antigen retrieval was performed by boiling the sections for 30 min in TEG buffer (pH = 9). Slides were washed in 50 mM $NM_4Cl$ medium for 30 min and then blocked with 1% BSA solution for 30 min. Anti-KIM-1 antibody (R&D Systems, #AF1817) was diluted (1:100) in 0.1% BSA, 0.3% Triton X-100. After overnight incubation at 4°C, slides were washed with 0.1% BSA and 0.3% Triton X-100 three times. Next, they were incubated with the corresponding secondary antibodies for 1 hr at room temperature. For immunohistochemistry on frozen kidney tissues, 10 µm sections from OCT blocks were thawed and permeabilized with 0.2% Triton X-100/PBS, then washed with PBS. Blocked with 2% BSA/PBS for 30 min, then sections were incubated with primary antibodies overnight at 4°C. Next day, the frozen sections were incubated with the respective secondary antibodies together with BODIPY 493/503 (Thermo Fisher Scientific, #D3922) for 1 hr at room temperature, then washed. As primary antibodies, goat anti-KIM-1 polyclonal antibody (1:100) (R&D Systems, #AF1817), rabbit anti-SGLT2 (1:200) (Abcam, #ab85626), and rabbit anti-megalin (1:1000) (courtesy of Prof. Michigami, Osaka Women's and Children's Hospital, Osaka, Japan) (*Yamagata et al., 2005*) were used. Immunofluorescence images were taken by TCS SP5 confocal microscope at a resolution of 2048 * 2048 pixels.

## Membrane packing measurement by C-Laurdan

Membrane packing measurements by C-Laurdan imaging were performed as described previously (*Kim et al., 2007*; *Levental et al., 2020*). Briefly, cultured cells were fixed in 4% paraformaldehyde for 10 min. After two washes in PBS, cells were stained with 10 µg/mL C-Laurdan (kindly provided by Dr. B.R. Cho, Korea University, South Korea) in PBS for 15 min at room temperature. Subsequently, cells were imaged at 37°C by a confocal microscope equipped with a multiphoton laser (Zeiss LSM 780 NLO). C-Laurdan was excited by the two-photon laser set to 800 nm. Two emission bands were acquired (band 1: 43–463 nm; band 2 473–503) using a ×20 objective. Next, individual pixels (containing the signals of the two bands) were trained and classified using Ilastik (v. 1.3.3) (*Berg et al., 2019*) as 'background,' 'nuclear,' 'perinuclear,' or 'periphery.' Of all pixels except 'background,' C-Laurdan GP values were calculated by the following formula: $GP = (I_{band1} - I_{band2})/(I_{band1} + I_{band2})$. For convenience, absolute GP values were *z*-scaled (mean = 0, SD = 1, over the full dataset). Images of pseudocolored GP values were generated in R (v. 4.1.0) using the packages ggplot2 (v. 3.3.5) and tiff (v. 0.1–8).

## Analysis of mRNA expression

Total RNA was isolated using the RNeasy Kit (74104, QIAGEN) including a DNAse treatment step. Concentration and purity of each sample were obtained from A260/A280 measurements in a micro-volume spectrophotometer NanoDrop-1000 (NanoDrop Technologies, Inc, Thermo Scientific). To measure the relative mRNA levels, quantitative (q)RT-PCR was performed using SYBR Green. cDNA was synthesized from 1 µg of total RNA with iScript cDNA Synthesis Kit (Bio-Rad, #1708891), following the manufacturer's instructions. The Power SYBR Green PCR Master Mix (Thermo Fisher Scientific, #4367659) was used for the PCR step. Amplification and detection were performed using the Mx3000P qPCR System (Agilent). Each mRNA from a single sample was measured in duplicate, using *18S* and *Actb* as housekeeping genes.

## Cytotoxicity assay

An IncuCyte S3 Live-Cell Analysis System was utilized to collect fluorescence and phase-contrast images over time. Cells were seeded at a density of 5000 cells per well in a 96-well culture microplate in the subconfluent experiments. For confluent experiments, 20,000 cells per well were seeded. 24 hr post-plating cells, IncuCyte Cytotox Red Reagent (EssenBio, #4632) was added to the medium (dilution 1:2000) together with BSA fatty acids. Cell confluence measurements and fluorescent objects were taken from four regions within each well, and values were averaged to calculate mean confluence per well or object counts per well. Results were represented as object counts normalized by confluence.

## Mitochondrial membrane potential assay

Cells were seeded at a density of 5000 cells per well in a 96-well culture microplate. 8 hr post-plating cells, BSA fatty acids treatment was performed. 24 hr post-plating cells, TMRE (Thermo Fisher

Scientific, # T669) was added to the medium (50 nM) for 15 min. Cells were washed twice with PBS, and full medium was added for the fluorescence measurement. Cells were immediately imaged in the red channel of IncuCyte S3 Live-Cell Analysis System. Results are represented as the image's average of the objects' mean fluorescent intensity.

### ROS detection assay

Cells were seeded at a density of 5000 cells per well in a 96-well culture microplate. 24 hr post-plating cells, DHE (Thermo Fisher Scientific, #D23107) was added to the medium (10 µM) together with BSA fatty acids. 4 hr after DHE addition, cells were imaged in the red channel of IncuCyte S3 Live-Cell Analysis System. Cell confluence measurements and fluorescent objects were taken from four regions within each well, and values were averaged to calculate mean confluence per well or object counts per well. Results are represented as objects counts normalized by confluence.

### Lipidomics

#### Lipid extraction for mass spectrometry lipidomics

Mass spectrometry-based lipid analysis was performed by Lipotype GmbH (Dresden, Germany) as described (*Sampaio et al., 2011*). Lipids were extracted using a two-step chloroform/methanol procedure (*Ejsing et al., 2009*). Samples were spiked with internal lipid standard mixture containing cardiolipin 14:0/14:0/14:0/14:0 (CL), ceramide 18:1;2/17:0 (Cer), diacylglycerol 17:0/17:0 (DAG), hexosylceramide 18:1;2/12:0 (HexCer), lyso-phosphatidate 17:0 (LPA), lyso-phosphatidylcholine 12:0 (LPC), lyso-phosphatidylethanolamine 17:1 (LPE), lyso-phosphatidylglycerol 17:1 (LPG), lyso-phosphatidylinositol 17:1 (LPI), lyso-phosphatidylserine 17:1 (LPS), phosphatidate 17:0/17:0 (PhA), phosphatidylcholine 17:0/17:0 (PC), phosphatidylethanolamine 17:0/17:0 (PE), phosphatidylglycerol 17:0/17:0 (PG), phosphatidylinositol 16:0/16:0 (PI), phosphatidylserine 17:0/17:0 (PS), cholesterol ester 20:0 (CE), sphingomyelin 18:1;2/12:0;0 (SM), sulfatide d18:1;2/12:0;0 (Sulf), triacylglycerol 17:0/17:0/17:0 (TAG), and cholesterol D6 (Chol). After extraction, the organic phase was transferred to an infusion plate and dried in a speed vacuum concentrator. First step dry extract was resuspended in 7.5 mM ammonium acetate in chloroform/methanol/propanol (1:2:4, V:V:V) and second step dry extract in 33% ethanol solution of methylamine in chloroform/methanol (0.003:5:1; V:V:V). All liquid handling steps were performed using Hamilton Robotics STARlet robotic platform with the anti-droplet control feature for organic solvents pipetting.

#### MS data acquisition

Samples were analyzed by direct infusion on a QExactive mass spectrometer (Thermo Scientific) equipped with a TriVersa NanoMate ion source (Advion Biosciences). Samples were analyzed in both positive and negative ion modes with a resolution of Rm/z = 200 = 280,000 for MS and Rm/z = 200 = 17,500 for MSMS experiments, in a single acquisition. MSMS was triggered by an inclusion list encompassing corresponding MS mass ranges scanned in 1 Da increments (*Surma et al., 2015*). Both MS and MSMS data were combined to monitor CE, DAG, and TAG ions as ammonium adducts; PC, PC O-, as acetate adducts; and CL, PA, PE, PE O-, PG, PI, and PS as deprotonated anions. MS only was used to monitor LPA, LPE, LPE O-, LPI, and LPS as deprotonated anions; Cer, HexCer, SM, LPC, and LPC O- as acetate adducts and cholesterol as ammonium adduct of an acetylated derivative (*Liebisch et al., 2006*).

#### Data analysis and postprocessing

Data were analyzed with in-house-developed lipid identification software based on LipidXplorer (*Herzog et al., 2012*; *Herzog et al., 2011*). Data postprocessing and normalization were performed using an in-house-developed data management system. Only lipid identifications with a signal-to-noise ratio > 5 and a signal intensity fivefold higher than in the corresponding blank samples were considered for further data analysis.

Lipidomics data are represented using the Cytoscape software (https://cytoscape.org).

## Intracellular FFA measurements

iRECs were seeded at a density of $1.5 \times 10^6$ in a 10 cm plate. Free fatty acids were measured by a colorimetric assay using the Free Fatty Acid Quantification Kit (Sigma-Aldrich, #MAK044) following the manufacturer's instructions.

## Statistics

Statistical analyses were performed using R Studio and GraphPad Prism. p-Value < 0.05 was considered significant (*p<0.05, **p<0.01, ***p<0.001, ****p<0.0001). More details on statistics can be found in each figure legend.

## Acknowledgements

We thank Sophie Berissi for help with histology staining procedures, Hermann-Josef Gröne for help with the histological evaluation, Shrey Kohli and Berend Iserman for help with urine analysis, and Yung-Hsin Shin and Gwenn Le Meur for excellent technical assistance. We acknowledge the European Research Council (ERC) under the European Horizon 2020 research and innovation programme (grant agreement no. 865408 [RENOPROTECT] to MS and no. 804474 [DiRECT] to SSL), the Deutsche Forschungsgemeinschaft (DFG SI1303/5-1 [Heisenberg-Programm] and DFG SI1303/6-1), the Steno Collaborative Grant from the NovoNordisk Foundation (NNF18OC0052457) (all to MS), and the Swiss National Science Foundation (NCCR Kidney.CH and 310030_189102) to SSL. AP-M was funded by a postdoctoral fellowship from the Fondation pour la Recherche Médicale (FRM-SPF20170938629).

## Additional information

### Competing interests

Julio Saez-Rodriguez: has received funding from GSK and Sanofi and consultant fees from Travere Therapeutics. The other authors declare that no competing interests exist.

### Funding

| Funder | Grant reference number | Author |
|---|---|---|
| European Research Council | 865408 | Jiayi Li |
| Novo Nordisk Foundation Center for Basic Metabolic Research | NNF18OC0052457 | Suresh Ramakrishnan |
| Deutsche Forschungsgemeinschaft | DFG SI1303/5-1 | Matias Simons |
| European Research Council | 804474 | Kelli Grand |
| Swiss National Centre of Competence in Research Kidney Control of Homeostasis | 310030_189102 | Soeren S Lienkamp |
| Fondation pour la Recherche Médicale | SPF20170938629 | Albert Pérez-Martí |

The funders had no role in study design, data collection and interpretation, or the decision to submit the work for publication.

### Author contributions

Albert Pérez-Martí, Conceptualization, Formal analysis, Investigation, Project administration, Visualization, Writing – original draft; Suresh Ramakrishnan, Jiayi Li, Luigi R De La Motte, Mélanie Parisot, Investigation; Aurelien Dugourd, Data curation, Formal analysis, Software, Visualization; Martijn R Molenaar, Formal analysis, Investigation, Visualization; Kelli Grand, Anis Mansouri, Formal analysis,

Software, Visualization; Soeren S Lienkamp, Julio Saez-Rodriguez, Funding acquisition, Resources; Matias Simons, Conceptualization, Funding acquisition, Project administration, Supervision, Writing – original draft

#### Author ORCIDs
Albert Pérez-Martí http://orcid.org/0000-0003-3234-3756
Aurelien Dugourd http://orcid.org/0000-0002-0714-028X
Martijn R Molenaar http://orcid.org/0000-0001-5221-608X
Matias Simons http://orcid.org/0000-0003-3959-6350

#### Ethics
All of the experimental protocols in this study were performed with the approval of the animal experimentation ethics committee of the University Paris Descartes (CEEA 34), projects registered as 17-058 and 20-022.

#### Decision letter and Author response
Decision letter https://doi.org/10.7554/eLife.74391.sa1
Author response https://doi.org/10.7554/eLife.74391.sa2

---

## Additional files

#### Supplementary files
• Transparent reporting form

#### Data availability
iRECs lipidomic data have been deposited in Dryad https://doi.org/10.5061/dryad.x95x69pm1. Kidney cortex of diabetic mice lipidomic data have been deposited in Dryad https://doi.org/10.5061/dryad.qv9s4mwgx. iRECs Transcriptome raw data (bam files) can be found at https://www.ncbi.nlm.nih.gov/sra/PRJNA809508. iRECs Transcriptome processed data (FPKM and DEG) have been deposited in Dryad DOI https://doi.org/10.5061/dryad.gqnk98sq7. The full code for the TF activity-lipid correlation analysis can be found in https://github.com/saezlab/Albert_perez_RNA_lipid/tree/main/scripts (copy archived at swh:1:rev:e445d92c5197dfc8afca02e47b680d89709efb7b).

The following datasets were generated:

| Author(s) | Year | Dataset title | Dataset URL | Database and Identifier |
|---|---|---|---|---|
| Pérez-Martí A | 2022 | Data from: Reducing lipid bilayer stress by monounsaturated fatty acids protects renal proximal tubules in diabetes | https://doi.org/10.5061/dryad.x95x69pm1 | Dryad Digital Repository, 10.5061/dryad.x95x69pm1 |
| Pérez-Martí A | 2022 | Data from: Reducing lipid bilayer stress by monounsaturated fatty acids protects renal proximal tubules in diabetes | https://doi.org/10.5061/dryad.qv9s4mwgx | Dryad Digital Repository, 10.5061/dryad.qv9s4mwgx |
| Pérez-Martí A | 2022 | Lipotoxicity in renal tubular epithelial cells | https://www.ncbi.nlm.nih.gov/bioproject/PRJNA809508 | NCBI BioProject, PRJNA809508 |
| Pérez-Martí A | 2022 | Data from: Reducing lipid bilayer stress by monounsaturated fatty acids protects renal proximal tubules in diabetes | https://doi.org/10.5061/dryad.gqnk98sq7 | Dryad Digital Repository, 10.5061/dryad.gqnk98sq7 |

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

# Appendix 1

**Appendix 1—table 1.** Nutrient composition of mouse diets.

|  | CTL DIET | MUFA-HFD | SFA-HFD |
|---|---|---|---|
| Macronutrients |  |  |  |
| Carbohydrates (kcal%) | 58 | 35 | 35 |
| Protein (kcal%) | 24 | 20 | 20 |
| Fat (kcal%) | 18 | 45 | 45 |
| Energy density (kcal/g) | 3.1 | 4.52 | 4.52 |
| Source of fat |  |  |  |
| Soybean Oil (kcal%) | 18 | 2.2 | 2.2 |
| Olive Oil (kcal%) | 0 | 42.8 | 0 |
| Butter, Anhydrous (kcal%) | 0 | 0 | 42.8 |
| Saturated | 15.5 | 14.3 | 62.6 |
| Monounsaturated | 23.9 | 69.5 | 30.7 |
| Polyunsaturated | 60.6 | 15.9 | 6.8 |
| Typical fatty acids composition (%) | Soybean Oil | Olive Oil | Butter, Anhydrous |
| C4, Butyric | 0 | 0 | 3.2 |
| C6, Caproic | 0 | 0 | 1.9 |
| C8 Caprylic | 0 | 0 | 1.1 |
| C10, Capric | 0 | 0 | 2.5 |
| C12, Lauric | 0 | 0 | 2.8 |
| C14, Myristic | 0.1 | 0 | 10 |
| C14:1, Myristoleic | 0 | 0 | 1.5 |
| C16, Palmitic | 10.4 | 11.5 | 26.2 |
| C16:1, Palmitoleic | 0.1 | 1.2 | 2.3 |
| C18, Stearic | 3.9 | 2.3 | 12.1 |
| C18:1, Oleic | 23 | 70.5 | 25.1 |
| C18:2, Linoleic | 51.8 | 13.0 | 2.3 |
| C18:3, Linolenic | 7.4 | 0.6 | 0 |
| C20, Arachidic | 0.4 | 0.4 | 1 |
| C20:1 | 0 | 0.2 | 0 |
| C22, Behenic | 0.3 | 0 | 0 |
| C24, Lignoceric | 0.2 | 0 | 0 |

**Appendix 1—table 2.** Spot urine parameters.

|  | Ctl Diet | Ctl Diet +STZ | MUFA-HFD+STZ | SFA-HFD+STZ |
|---|---|---|---|---|
| UACR (µg/mg) | 28.71±8.30 | 64.81±38.28 | 177±59.10 **** # | 97.41±21.76* |
| Glucose / Creatinine (mg/mg) | 0.8599±0.34 | 2734±1360*** | 1433±859.90* | 1435±821.10* |

Urine albumin-to-creatinine ratio (UACR) and Urine glucose-to-creatinine ratio at 14 weeks after STZ injection. Data are presented as mean ± StDev. *p<0.05, ****p<0.0001 vs Ctl Diet; #p<0.05 vs Ctl Diet + STZ (Kruskal-Wallis plus Dunn's multiple comparisons test). n=7 Ctl diet, n=8 Ctl Diet + STZ, n=8 MUFA-HFD + STZ, n=7 SFA-HFD + STZ.

