## [Editor Report]

This study addresses the differential effect of saturated versus monounsaturated lipids on damage to kidney tubules. The authors find that the majority of the lipids are taken up in the S2 and S3 segments of the proximal tubule. They find that while the unsaturated fatty acid increased lipid storage in the tubule cells, it produced less cell damage than the saturated fatty acid due to decreased endoplasmic reticulum stress. These studies emphasize a potential beneficial effect of a diet rich in monounsaturated fatty acids, especially in patients with chronic kidney disease.

---

## [Decision Letter]

**Decision letter after peer review:**

Thank you for submitting your article "Increasing triacylglycerol formation and lipid storage by unsaturated lipids protects renal proximal tubules in diabetes" for consideration by *eLife*. Your article has been reviewed by 3 peer reviewers, one of whom is a member of our Board of Reviewing Editors, and the evaluation has been overseen by Martin Pollak as the Senior Editor. The reviewers have opted to remain anonymous.

The study addresses the role of different lipids on renal damage in DKD and demonstrate differences in saturated versus mono-unsaturated lipids on tubular damage in a mouse model of DKD. These diet models are combined with transcriptomic, lipidomics, and in vitro analyses, including the analyses of mitochondrial effects. The topic is timely and provides new insights into the effects of lipids on DKD. However, there remain concerns that the mechanistic studies were only done in vitro and there is limited attempt to correlate with in vivo observations. A number of other issues were raised that need to be addressed before the manuscript can be considered further for possible publication.

There needs to be a clear identification of nephron segments accumulating the lipid droplets. Further experiments are required to quantitate and characterize the lipid accumulation and definitively identify the nephron segments accumulating lipid droplets. Although discussion suggests possible differences for lipid accumulation in S1, S2 and S3 segments, there is no demonstration of specific segments of the nephron provided In addition, since there was not evidence of altered uptake in the S1 segment, does this indicate that the increased uptake in vivo is predominantly basolateral rather than through fatty acid bound albumin? The authors should examine altered expression of fatty acid binding proteins such as FATP and CD36.

There remains uncertainty about the mechanism by which MUFA inhibit injury by simultaneous addition of SFA. although the authors postulate it is due in part to increased TGA formation, the protective effect remained even with dacylglycerol transferase inhibition.

Existing literature provides strong support that saturated fatty acids, in particular palmitate, are associated with lipotoxicity, whereas unsaturated lipids such, as oleate, are protective against palmitate toxicity. Oleate protection is thought to promote triacylglycerol formation from palmitate preventing its metabolism and being a preferred substrate for DGAT1 and DGAT2. The manuscript describes several of the established cellular effects of SFA and MUFA. The authors should describe the current understanding of the different pathways for palmitate and oleate metabolism leading to their distinct effects and clearly identify the new and different pathways for palmitate and oleate metabolism in the kidney.

Existing literature provides strong support that saturated fatty acids, in particular palmitate, are associated with lipotoxicity, whereas unsaturated lipids such, as oleate, are protective against palmitate toxicity. Oleate protection is thought to promote triacylglycerol formation from palmitate preventing its metabolism and being a preferred substrate for DGAT1 and DGAT2. The manuscript describes several of the established cellular effects of SFA and MUFA. The authors should describe the current understanding of the different pathways for palmitate and oleate metabolism leading to their distinct effects and clearly identify the new and different pathways for palmitate and oleate metabolism in the kidney.

While the design of the studies uses dietary manipulation to assess effects of butter and olive oil rich diets to study renal response to injury, these results should be put in the context of inconsistent evidence supporting the hypothesis that dietary change in proteins, lipids, nutritional supplements etc. alter kidney disease.

The authors also need to address all of the points raised in the reviews.

*Reviewer #1 (Recommendations for the authors):*

Pérez-Martí address in their manuscript the role of different lipids on renal damage in DKD. They use different diet composition to address the role of saturated versus mono-unsaturated lipids on tubular damage in a mouse model of DKD. These diet models are combined with transcriptomic, lipidomics, and in vitro analyses, including the analyses of mitochondrial effects. The topic is timely and the data are of interest, as they provide new insights into the effects of lipids on DKD.

– In the abstract the authors write that it is unclear whether LDs are protective or damaging in DKD; in the introduction the authors later state that the concept of lipidtoxicity is well established – which I would agree upon. The authors need to rephrase the statement in the abstract – isn't a key aspect the comparison of SFA and MUFAs?

– Was the weight of organs (in particular of the kidney) normalized for the body weight?

– Page 4: if MUFA-HFD favored accumulation of LDs in the kidney – what is the "fate" at later timepoints? It is hard to imagine that their continues presence will not be harmful as well. The authors themselves raise this point later in the discussion and should extent on this aspect.

– Page 5: when incubating cells in vitro with PA or OA, was the same phenotype as in vivo observed (e.g. accumulation of LDs upon OA exposure)? Or are there differences between a long-term exposure in vivo versus a short-term exposure in vitro?

– Page 6: the authors conclude that the TF analyses confirmed the initial transcriptomic data analysis. Is this not a moot point? The transcriptomic data were used for the TF analyses – hence everything else would be a surprise. I would suggest rephrasing.

– Page 6: to study the role of the different ER-stress response arms, the authors use inhibitors. To inhibit IRE1a (should be "α", not a in the text) they used 4µ8C, which is an inhibitor of IRE1a's RNAse activity. Hence, a role of the IRE1a response arm has not been fully addressed.

– The observation made with tunicamycin and thapsigargin are of interest. But what is their relevance in the context of the current story?

– Page 8: did the authors also measure cardiolipines and their oxidation? This would strengthen the results obtained in regard to the mitochondrial phenotype and increased ROS generation.

– Page 8: were the authors able to confirm changes observed in lipid composition in vitro also in vivo?

– In general, it is a limitation that the mechanistic studies have been done in vitro. This is a limitation for the conclusions drawn. This needs to be acknowledged in the discussion.

– I am surprised that spot urine was used. Spot urine values show large variations and metabolic cages are commonly used to avoid this. I do not share the authors concern.

*Reviewer #2 (Recommendations for the authors):*

This study examined the potential role of unsaturated fatty acids (MUFA) vs. saturated fatty acids (SFA) in mediation of kidney proximal tubule injury in a model of diabetes. Utilizing cultured proximal tubule cells, they find that MUFA led to increased tubule lipid droplets by stimulating triacylglycerol formation and prevented injury by decreasing ER stress compared to SFA. Of note simultaneous administration of MUFA still inhibited the effect of SFA. In general, the results appear consistent and the studies appear to be well performed.

There remains some uncertainty about the mechanism by which MUFA inhibit injury by simultaneous addition of SFA. although the authors postulate it is due in part to increased TGA formation, the protective effect remained even with dacylglycerol transferase inhibition. The authors may wish to speculate further.

Since there was not evidence of altered uptake in the S1 segment, does this indicate that the increased uptake in vivo is predominantly basolateral rather than through fatty acid bound albumin? The authors should examine altered expression of fatty acid binding proteins such as FATP and CD36>

Other questions:

Was there any significant difference in UACR between SFA and MUFA since MUFA appears to be higher?

The studies with autophagy are incomplete since the increased p62 suggests ineffective autophagy.

Is the PERK inhibition with 5 µM GSK206414 significantly different than the DMSO group?

line 144 This should be Figure 1G.

*Reviewer #3 (Recommendations for the authors):*

The central tenet of this investigation that accumulation of lipid droplets within the kidney tubules is beneficial in kidney injury should be supported with additional studies. These experiments should clearly quantitate and characterize the lipid accumulation and definitively identify the nephron segments accumulating lipid droplets. Although discussion suggests possible differences for lipid accumulation in S1, S2 and S3 segments, there is no demonstration of specific segments of the nephron provided.

The conclusion that LD are protective would be strengthened by concurrent assessment in lipid accumulation and cellular injury in the disease setting (e.g., KIM1). This seems especially pertinent in view of the heterogeneity in lipid accumulation among the tubular segments.

As presented, the in vitro studies document many of the expected pathways for cellular handling of palmitate and oleate. The novelty of the observations concern that these pathways are occurring in kidney cells in a disease setting. At present there is no adjustment in the design of the cell culture studies that accounts for renal cell handling of these fatty acids in the diabetic setting which may include the effects of glucose on lipid metabolism.

Existing literature provides strong support that saturated fatty acids, in particular palmitate, are associated with lipotoxicity, whereas unsaturated lipids such, as oleate, are protective against palmitate toxicity. Oleate protection is thought to promote triacylglycerol formation from palmitate preventing its metabolism and being a preferred substrate for DGAT1 and DGAT2. The manuscript describes several of the established cellular effects of SFA and MUFA. The authors should describe the current understanding of the different pathways for palmitate and oleate metabolism leading to their distinct effects and clearly identify the new and different pathways for palmitate and oleate metabolism in the kidney.

While the design of the studies uses dietary manipulation to assess effects of butter and olive oil rich diets to study renal response to injury, these results should be put in the context of inconsistent evidence supporting the hypothesis that dietary change in proteins, lipids, nutritional supplements etc. alter kidney disease.

---

## [Author Response]

There needs to be a clear identification of nephron segments accumulating the lipid droplets. Further experiments are required to quantitate and characterize the lipid accumulation and definitively identify the nephron segments accumulating lipid droplets. Although discussion suggests possible differences for lipid accumulation in S1, S2 and S3 segments, there is no demonstration of specific segments of the nephron provided.

Thank you for this important comment. By using immunohistochemistry on the four different conditions (control, STZ, MUFA-STZ and SFA-STZ) we have now indeed confirmed that lipid droplet (LD) accumulation occurs mainly in S2 and S3 segments. To label LDs we used BODIPY, while segments were marked with either megalin for all three PT segments or SGLT2 for only the S1 segment. The results provide strong evidence for LD accumulation within proximal tubular cells of the S2/S3 segments upon high fat diet treatment. Interestingly, we also found that the damage marker KIM-1 was found in cells distinct from these cells, confirming our cell culture findings that cells with TAG and LD formation protects against lipotoxic injury. These new important data are now presented in Figure1—figure supplement 2B and Figure 8B,C.

In addition, since there was not evidence of altered uptake in the S1 segment, does this indicate that the increased uptake in vivo is predominantly basolateral rather than through fatty acid bound albumin? The authors should examine altered expression of fatty acid binding proteins such as FATP and CD36.

As requested, we have now performed expression analysis of the apical FATP2 and the basolateral CD36 by qPCR. The data show that both FATP and CD36 are strongly downregulated in all three mice groups treated with STZ compared to the control group (new Figure1—figure supplement 2). This result may argue against a role of FATP2 and CD36 in the observed differences in LDs accumulation between groups. We therefore suspect that LD accumulation in S2/S3 segments is caused by enhanced capacity for TAG formation, potentially due to the reported higher levels of the desaturase SCD1 in these segments. Alternatively, it may be that the enhanced mitochondrial β-oxidation in S1 segments prevents lipid storage.

There remains uncertainty about the mechanism by which MUFA inhibit injury by simultaneous addition of SFA. although the authors postulate it is due in part to increased TGA formation, the protective effect remained even with dacylglycerol transferase inhibition.

The observation that the protective effect of OA remained upon DGAT inhibition may suggest that OA co-treatment can balance the saturation level of ER phospholipids even when TAG and LD formation is blocked. In fact, our lipidomics analysis could show that in OA/PA-treated cells DGAT inhibition increases DAG levels (at the expense of TAG) but does not change the saturation level of DAG. In other words, OA co-treatment can reduce di-saturated DAG levels, especially 16:0_16:0, compared to PA alone no matter if DGAT is inhibited or not. Consequently, there is also more 16:0_18:1 that does not harm the ER membrane as 16:0_16:0 is able to do.

In the revised version, we have attempted to explain the DGAT inhibition results in more detail (p.10, line 14-22; p.13, line 18-23) and we are also showing the abovementioned DAG profile in Figure 7—figure supplement 2.

Existing literature provides strong support that saturated fatty acids, in particular palmitate, are associated with lipotoxicity, whereas unsaturated lipids such, as oleate, are protective against palmitate toxicity. Oleate protection is thought to promote triacylglycerol formation from palmitate preventing its metabolism and being a preferred substrate for DGAT1 and DGAT2. The manuscript describes several of the established cellular effects of SFA and MUFA. The authors should describe the current understanding of the different pathways for palmitate and oleate metabolism leading to their distinct effects and clearly identify the new and different pathways for palmitate and oleate metabolism in the kidney.

Our manuscript indeed describes some of the known effects of PA and OA but for the first time this is put into the context of proximal tubular pathophysiology in DKD. We provide a comprehensive and mechanistic explanation of the toxic and protective effects of PA and OA, respectively. Moreover, we highlight the differential effects of excess lipids on the different PTC segments that should stimulate follow-up studies about the differences in lipid uptake and metabolism between the segments. In the revised version, we have attempted to highlight the novel pathways for MUFA/SFA metabolism in the kidney (e.g. p. 3 , line 23-24 and p. 13, line 31-37). We now also provide more references for the damaging effects of SFAs in the introduction (please see p. 3, line 15-22).

Existing literature provides strong support that saturated fatty acids, in particular palmitate, are associated with lipotoxicity, whereas unsaturated lipids such, as oleate, are protective against palmitate toxicity. Oleate protection is thought to promote triacylglycerol formation from palmitate preventing its metabolism and being a preferred substrate for DGAT1 and DGAT2. The manuscript describes several of the established cellular effects of SFA and MUFA. The authors should describe the current understanding of the different pathways for palmitate and oleate metabolism leading to their distinct effects and clearly identify the new and different pathways for palmitate and oleate metabolism in the kidney.While the design of the studies uses dietary manipulation to assess effects of butter and olive oil rich diets to study renal response to injury, these results should be put in the context of inconsistent evidence supporting the hypothesis that dietary change in proteins, lipids, nutritional supplements etc. alter kidney disease.

In fact, the first paragraph of the discussion is already dedicated to the current knowledge about the effects of Mediterranian diets on kidney health. To put this into the general context of DKD management, however, we have now added the dietary recommendations of the Kidney Disease: Improving Global Outcomes (KDIGO) initiative from 2020 (p.11, line 36- p.12 line 4). It is of interest that in these important guidelines, lipids do not play a major role. Clearly, more emphasis was placed on low protein and low sodium, which probably reflects the limited knowledge on the impact of fat for kidney health.

The authors also need to address all of the points raised in the reviews.Reviewer #1 (Recommendations for the authors):Pérez-Martí address in their manuscript the role of different lipids on renal damage in DKD. They use different diet composition to address the role of saturated versus mono-unsaturated lipids on tubular damage in a mouse model of DKD. These diet models are combined with transcriptomic, lipidomics, and in vitro analyses, including the analyses of mitochondrial effects. The topic is timely and the data are of interest, as they provide new insights into the effects of lipids on DKD.– In the abstract the authors write that it is unclear whether LDs are protective or damaging in DKD; in the introduction the authors later state that the concept of lipidtoxicity is well established – which I would agree upon. The authors need to rephrase the statement in the abstract – isn't a key aspect the comparison of SFA and MUFAs?

Thank you for this point. We changed title and abstract accordingly.

– Was the weight of organs (in particular of the kidney) normalized for the body weight?

We prefer to show the absolute values because the greater loss of WAT and muscle (as suggested by heart weight) in the control Diet + STZ group would give a higher relative weight for the kidneys of this group.

**Author response image 1. sa2fig1:** Tissue weight normalized for body weight. Data are presented as mean ± SEM. *p<0.05, **p<0.01, ****p<0.0001; One-way ANOVA plus Holm-Sidak’s multiple comparisons test. n=7 Ctl diet, n=8 Ctl Diet + STZ, n=8 MUFA-HFD + STZ, n=7 SFA-HFD + STZ.

– Page 4: if MUFA-HFD favored accumulation of LDs in the kidney – what is the "fate" at later timepoints? It is hard to imagine that their continues presence will not be harmful as well. The authors themselves raise this point later in the discussion and should extent on this aspect.

Thank you for this comment. In the mouse, we stopped the experiments at 20 weeks. This experiment cannot be prolonged for this paper, which is why we acknowledged this limitation in the discussion. What we can conclude from this 20 week-experiment, however, is that – at least based on KIM-1 expression – PTCs accumulating LDs show less damage than PTCs. This result may suggest that LDs confer long-term protection also in vivo.

Following the advice of the referee, we have now performed OA treatment for 3 days and 7 days in iRECs. We used the same concentration as for PA (leading to cell death within 20 h). Even after 7 days, we did not detect any decrease in cell viability in OA-treated cells, confirming the benign effects of OA. These new results can now be found in Figure 2—figure supplement 1B.

– Page 5: when incubating cells in vitro with PA or OA, was the same phenotype as in vivo observed (e.g. accumulation of LDs upon OA exposure)? Or are there differences between a long-term exposure in vivo versus a short-term exposure in vitro?

We have now also performed in vivo lipidomics. Despite the temporal differences between in vitro and in vivo lipid treatments, we observed a striking similarity with regard to the LD stainings and in the lipidomic profiles. While there were more LD in OA- treated cells and MUFA-HFD fed mice, TAG synthesis was impaired in SFA-HFD fed mice in a similar way as in the PA-treated cells. Additionally, we now show that damage mostly occurs in PTCs with less LDs, which correlates with the protective effect of OA and LDs in cultured cells.

– Page 6: the authors conclude that the TF analyses confirmed the initial transcriptomic data analysis. Is this not a moot point? The transcriptomic data were used for the TF analyses – hence everything else would be a surprise. I would suggest rephrasing.

Thank you. We have now rephrased this conclusion.

– Page 6: to study the role of the different ER-stress response arms, the authors use inhibitors. To inhibit IRE1a (should be "α", not a in the text) they used 4µ8C, which is an inhibitor of IRE1a's RNAse activity. Hence, a role of the IRE1a response arm has not been fully addressed.

Thanks for this comment. We have now used the chemical inhibitor GSK2850163, which inhibits both the kinase and the RNase activity. Interestingly the use of this inhibitor proved that the IRE1alpha branch exerts a protective effect against PA. These new results can be found in Figure 3B.

– The observation made with tunicamycin and thapsigargin are of interest. But what is their relevance in the context of the current story?

We used these inhibitors to demonstrate that protective effects of OA are specific for PA-induced ER stress. This is important because it could otherwise be argued that lipid bilayer stress is secondary to other ER stressors, such as misfolded proteins. We added a sentence to emphasize this conclusion (p.7, line 28-30).

– Page 8: did the authors also measure cardiolipines and their oxidation? This would strengthen the results obtained in regard to the mitochondrial phenotype and increased ROS generation.

Following the advice of the referee, we have now rechecked our lipidomics data, and we could indeed detect a strong effect of OA on cardiolipin formation. In particular, we see that cardiolipin levels are decreased in IRECs treated with OA, possibly because they share precursors with TAG biosynthesis pathway (Author response image 2). Moreover, while we could not detect cardiolipins with less than 4 double bounds in OA treated cells, PA treatment increased the amount of cardiolipin species with only two double bounds. These findings are interesting because they could partly be responsible for the lower mitochondrial membrane potential observed in iRECs treated with OA. Yet, we decided not to include the cardiolipins in the manuscript due to the length restrictions and our focus on ER phospholipids.

**Author response image 2. sa2fig2:** Cardiolipin profile of iRECs treated with PA, OA and PA/OA (A) Cardiolipin levels in iRECs treated for 16h with BSA, PA 250µM, OA 250µM and PA 250µM plus OA 125µM. Data are presented as mean ± SEM; **p<0.01; One-way ANOVA and Holm-Sidak’s multiple comparisons test; n=3. (B) Relative levels of lipid classes involved in the synthesis of TAGs and cardiolipins in iRECs treated for 16h with BSA and OA 250µM. The size of the circles is set to the arbitrary unit of 1 for the BSA cells. Lipid classes not identified are shown in grey. LPA: lyso-phosphatidic acids; PhA: phosphatidic acids; DAG: diacylglycerol; TAG: triacylglycerol; PG: phosphatidylglycerol; CDP: cytidine diphosphate. (n=3) (C) Relative amount of Cardiolipin species classified by the number of double bonds in iRECs treated for 16h with BSA, PA 250µM, OA 250µM and PA 250µM plus OA 125µM. Data are presented as mean ± SEM ; n=3.

– Page 8: were the authors able to confirm changes observed in lipid composition in vitro also in vivo?

Thank you for this important point. We have now performed shotgun lipidomics on the renal cortices of all four mouse groups. As stated above, the data show a striking similarity with the cell culture lipidomics data, despite the fact that the type and duration of HFD treatment differs vastly between the two models. In particular, we were able to confirm our key finding namely that monounsaturated lipids promote TAG synthesis, while saturated lipids cause accumulation of di-saturated TAG precursors. These results can be found in the new Figure 8A.

– In general, it is a limitation that the mechanistic studies have been done in vitro. This is a limitation for the conclusions drawn. This needs to be acknowledged in the discussion.

The revised version has overcome this limitation. On the one hand, we performed lipidomics on mouse renal cortical samples. The obtained lipidomes for MUFA-STZ and SUFA-STZ mice are indeed very similar to the OA and PA treatment on cultured cells despite the differences in the length of the treatment. In particular, we were able to show that SFA also leads to the accumulation of di-saturated DAGs. On the other hand, we found that KIM-1 positive cells in the renal cortex are mostly devoid of LDs, suggesting that TAG and LD formation may also be protective in vivo.

– I am surprised that spot urine was used. Spot urine values show large variations and metabolic cages are commonly used to avoid this. I do not share the authors concern.

In the approval by the local Animal Ethics Committee for our mouse work, one of the defined humane endpoints was a weight loss that is bigger than a 20%. As mice usually reduce the food intake during the habituation to the metabolic cages, we decided not to use metabolic cages to preserve enough mice in the study, given that some of the mice were already near the endpoint. It is indeed possible that this might have had some impact on the quality of the urine samples.

Reviewer #2 (Recommendations for the authors):This study examined the potential role of unsaturated fatty acids (MUFA) vs. saturated fatty acids (SFA) in mediation of kidney proximal tubule injury in a model of diabetes. Utilizing cultured proximal tubule cells, they find that MUFA led to increased tubule lipid droplets by stimulating triacylglycerol formation and prevented injury by decreasing ER stress compared to SFA. Of note simultaneous administration of MUFA still inhibited the effect of SFA. In general, the results appear consistent and the studies appear to be well performed.There remains some uncertainty about the mechanism by which MUFA inhibit injury by simultaneous addition of SFA. although the authors postulate it is due in part to increased TGA formation, the protective effect remained even with dacylglycerol transferase inhibition. The authors may wish to speculate further.

As stated above, we believe that this result demonstrates that OA co-treatment can balance the saturation level of ER phospholipids even if LD formation is blocked by DGAT inhibition. In fact, our lipidomics analysis could show that OA co-treatment can reduce di-saturated DAG levels, especially 16:0_16:0, compared to PA alone no matter if DGAT is inhibited or not. As a result, there is more 16:0_18:1 that does not harm the ER membrane as 16:0_16:0 is able to do. These data can now be found in Figure 7—figure supplement 2. We also added a passage in the discussion about this point (p. 13, line 18-23), and we changed title and abstract to put more emphasis on ER membrane saturation than on LDs.

Since there was not evidence of altered uptake in the S1 segment, does this indicate that the increased uptake in vivo is predominantly basolateral rather than through fatty acid bound albumin? The authors should examine altered expression of fatty acid binding proteins such as FATP and CD36>

We repeat here our response to the same comment from the editor: As requested, we have now performed expression analysis of the apical FATP2 and the basolateral CD36 by qPCR. The data show that both FATP and CD36 are strongly downregulated in all three mice groups treated with STZ compared to the control group (new Figure 1—figure supplement 2). This result may argue against any role of FATP2 and CD36 in the observed differences in LDs accumulation between groups. We therefore suspect that LD accumulation in S2/S3 segments is caused by an enhanced capacity for TAG formation, potentially due to the reported higher levels of the desaturase SCD1 in these segments. Alternatively, it may be that the enhanced mitochondrial β-oxidation in S1 segments prevents lipid storage.

Other questions:Was there any significant difference in UACR between SFA and MUFA since MUFA appears to be higher?

No, there is not a significant difference. Despite a visible trend, the data did not pass the normality test.

The studies with autophagy are incomplete since the increased p62 suggests ineffective autophagy.

Thank you for this point. We have now performed a more thorough autophagy analysis by using Western Blotting of p62 and LC3 in combination with Bafilomycin A1 treatment. The results suggest that autophagic flux is preserved but autophagic cargo accumulates, most likely due to an excess of autophagic cargo (e.g. misfolded proteins due to ER stress). These new results can be found in Figure 3—figure supplement 1B-D.

Is the PERK inhibition with 5 µM GSK206414 significantly different than the DMSO group?

Sorry about this. There was a mistake in the figure, the asterisk was supposed to mark the difference between DMSO and 5 µM GSK206414. They are significantly different.

line 144 This should be Figure 1G.

Thanks a lot. We have now changed this.

Reviewer #3 (Recommendations for the authors):The central tenet of this investigation that accumulation of lipid droplets within the kidney tubules is beneficial in kidney injury should be supported with additional studies. These experiments should clearly quantitate and characterize the lipid accumulation and definitively identify the nephron segments accumulating lipid droplets. Although discussion suggests possible differences for lipid accumulation in S1, S2 and S3 segments, there is no demonstration of specific segments of the nephron provided.

By using immunohistochemistry on the four different conditions (control, STZ, MUFA-STZ and SFA-STZ) we have now confirmed that LD accumulation occurs mainly in the S2 and S3 segments. To label LDs we used BODIPY, segments were marked with either megalin for all three PT segments or SGLT2 for only the S1 segment. Altogether, the findings provide strong evidence for LD accumulation within proximal tubular cells of the S2/S3 segments upon high fat diet treatment. See also our comments above (first point of editor).

The conclusion that LD are protective would be strengthened by concurrent assessment in lipid accumulation and cellular injury in the disease setting (e.g., KIM1). This seems especially pertinent in view of the heterogeneity in lipid accumulation among the tubular segments.

Thank you for raising this very important issue. We have now performed additional immunohistochemistry. They show that KIM-1 immunoreactivity is mainly seen in PTCs with no or less LDs. By contrast, KIM-1 immunoreactivity overlapped with markers for the S1 segment. Due to the more widespread expression of these damage markers in SFA-STZ mice this colocalization was more evident in these mice. Altogether, these results provide strong in vivo-validation for the protective effects of TAG and LD formation.

As presented, the in vitro studies document many of the expected pathways for cellular handling of palmitate and oleate. The novelty of the observations concern that these pathways are occurring in kidney cells in a disease setting. At present there is no adjustment in the design of the cell culture studies that accounts for renal cell handling of these fatty acids in the diabetic setting which may include the effects of glucose on lipid metabolism.

The regular medium for iRECs culture is indeed DMEM with high glucose (4.5mg/mL). This concentration is comparable to the plasma glucose concentration in diabetic mice. Following the reviewer’s suggestion, we have now additionally studied the effect of a stepwise glucose increase for PA-induced cytotoxicity. As can be seen in Figure 2—figure supplement 1C, increasing glucose concentrations showed a visible but non-significant enhancement of the cytotoxic effect of PA.

Existing literature provides strong support that saturated fatty acids, in particular palmitate, are associated with lipotoxicity, whereas unsaturated lipids such, as oleate, are protective against palmitate toxicity. Oleate protection is thought to promote triacylglycerol formation from palmitate preventing its metabolism and being a preferred substrate for DGAT1 and DGAT2. The manuscript describes several of the established cellular effects of SFA and MUFA. The authors should describe the current understanding of the different pathways for palmitate and oleate metabolism leading to their distinct effects and clearly identify the new and different pathways for palmitate and oleate metabolism in the kidney.

We repeat here our response to the same comment from the editor: Our manuscript indeed describes some of the known effects of PA and OA but for the first time this is put into the context of proximal tubular pathophysiology in DKD. We provide a comprehensive and mechanistic explanation of the toxic and protective effects of PA and OA, respectively. Moreover, we highlight the differential effects of excess lipids on the different PTC segments that should stimulate follow-up studies about the differences in lipid uptake and metabolism between the segments. In the revised version, we have attempted to highlight the novel pathways for MUFA/SFA metabolism in the kidney (e.g. p. 3 , line 23-24 and p. 13, line 31-37). We now also provide more references for the damaging effects of SFAs in the introduction (please see p. 3, line 15-22).

While the design of the studies uses dietary manipulation to assess effects of butter and olive oil rich diets to study renal response to injury, these results should be put in the context of inconsistent evidence supporting the hypothesis that dietary change in proteins, lipids, nutritional supplements etc. alter kidney disease.

We repeat here our response to the same comment from the editor: In fact, the first paragraph of the discussion is already dedicated to the current knowledge about the effects of Mediterranean diets on kidney health. To put this into the general context of DKD management, however, we have now added the dietary recommendations of the Kidney Disease: Improving Global Outcomes (KDIGO) initiative from 2020 (p.11, line 36- p.12 line 4). It is of interest that in these important guidelines, the fat type does not play a major role. Clearly, more emphasis was placed on low protein and low sodium, which probably reflects the limited knowledge on the impact of unsaturated fat for kidney health.